# Generalized Kernelized Bandits: Self-Normalized Bernstein-Like Dimension-Free Inequality and Regret Bounds

## Abstract

We study the regret minimization problem in the novel setting of *generalized kernelized bandits* (GKBs), where we optimize an unknown function $f^*$ belonging to a *reproducing kernel Hilbert space* (RKHS) having access to samples generated by an *exponential family* (EF) noise model whose mean is a non-linear function $\mu(f^*)$. This model extends both *kernelized bandits* (KBs) and *generalized linear bandits* (GLBs). We propose an optimistic algorithm, `GKB-UCB`, and we explain why existing self-normalized concentration inequalities do not allow to provide tight regret guarantees. For this reason, we devise a novel self-normalized Bernstein-like dimension-free inequality resorting to Freedman's inequality and a stitching argument, which represents a contribution of independent interest. Based on it, we conduct a regret analysis of `GKB-UCB`, deriving a regret bound of order $\widetilde{O}(\gamma_T \sqrt{T/\kappa_*})$, being $T$ the learning horizon, $\gamma_T$ the maximal information gain, and $\kappa_*$ a term characterizing the magnitude the reward nonlinearity. Our result matches, up to multiplicative constants and logarithmic terms, the state-of-the-art bounds for both KBs and GLBs and provides a *unified view* of both settings.

## 1 Introduction

*Multi-Armed Bandits* [MABs, 15] have been extensively studied and extended over the years. One key research direction involves expanding the MAB framework to continuous action spaces. Doing this requires introducing some notion of similarity or structure in the expected rewards relative to the distance between arms. Without such structure, information gathered from explored actions/arms cannot be transferred to unexplored ones, making learning infeasible [4]. The most known and studied structure over the arms is the *linear* one, and led to the design of *linear bandits* [LBs, 1, 6]. In LBs, the expected reward is modeled as the inner product between the action and an unknown parameter vector (i.e., $\mathbb{E}[y_t|\mathbf{x}_t; \boldsymbol{\theta}^*] = \langle \mathbf{x}_t, \boldsymbol{\theta}^* \rangle$). This setting strictly generalizes the finite-arms MABs [15, 23] that can be retrieved considering arms as in an $\mathbb{R}^d$ canonical basis.

LBs, in turn, have been extended in parallel in two directions: *generalized linear bandits* [GLBs, 10] and *kernelized bandits* [KBs, 5, 29]. On the one hand, GLBs employ a *generalized linear model* [GLM, 19] to allow for the representation of different noise models (including Gaussian and Bernoulli). This is achieved with the use of a real-valued non-linear *inverse link function* $\mu(\cdot)$, such that the expected payoff is defined as $\mathbb{E}[y_t|\mathbf{x}_t; \boldsymbol{\theta}^*] = \mu(\langle \mathbf{x}_t, \boldsymbol{\theta}^* \rangle)$. On the other hand, KBs focus on the optimization of an unknown expected reward function belonging to a *reproducing kernel Hilbert space* (RKHS) induced by a known kernel function $k(\mathbf{x}, \mathbf{x}')$, often resorting to Gaussian processes for designing algorithms [22]. We observe that GLBs fall back to LBs when the identity link function $\mu = I$ is considered, and KBs fall back to LBs when a linear kernel $k(\mathbf{x}, \mathbf{x}') = \langle \mathbf{x}, \mathbf{x}' \rangle$ is considered.

Submitted to 39th Conference on Neural Information Processing Systems (NeurIPS 2025). Do not distribute.

In this work, we propose the novel *generalized kernelized bandit* (GKB) setting, which unifies GLBs and KBs (Figure 1). This setting enables learning in the scenarios in which the unknown function $f^*$ comes from an RKHS and the samples come from an exponential family model whose mean is obtained by applying an inverse link function $\mu$ to function $f^*$. This allows accounting for a variety of noise models, including Gaussian and Bernoulli [3].

As established by the literature [1, 9, 17], when designing *optimistic* regret minimization algorithms for either GLBs and KBs, a fundamental technical tool are *self-normalized* concentration inequalities [7]. When targeting regret minimization in the novel setting of GKBs, it is necessary to employ a concentration inequality that combines the requirements of GLBs and KBs, i.e., it should avoid dependencies on the minimum slope $\dot{\mu}$ of the inverse link function (as in GLBs) and on the dimensionality of the feature representation (as in KBs). The seminal work [1] provides a self-normalized concentration inequality for least square estimators under subgaussian noise, exploit-

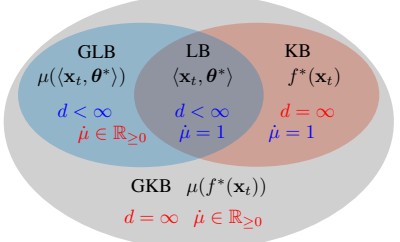

Figure 1: Inclusion of the settings ($f(\cdot)$ is assumed to belong to a RKHS).

ing theoretical advancements in self-normalized processes and pseudo-maximization of [7, 8]. However, this inequality does not conveniently manage the case in which the samples come from an exponential family model where the variances depend on inverse link function $\mu$, ultimately leading to a dependence on its minimum slope. To cope with this issue, [9] derive a concentration inequality via a pseudo-maximization technique that results in a tight regret bound for GLBs, accounting for the heteroscedastic characteristics of the noise (i.e., Bernstein-like). However, their concentration inequality presents a dependency on the dimensionality of the feature vector (i.e., dimension-dependent). While not being problematic for GLBs, this hinders a direct application to GKBs, where the feature representation (induced by the kernel function) can be infinite-dimensional. Additionally, [5] design a self-normalized bound for martingales which provides tight concentration results for the KB setting, directly operating with kernels. However, this result can be considered the counterpart of [1] in the dual (kernel) space and, for this reason, it shares the same limitation when using an inverse link function, generating a dependence on the minimum value of $\dot{\mu}$ when applied to GKBs.[1] It appears now necessary to derive a novel concentration result that is both dimension-free and Bernstein-like to properly address the GKB setting.

**Outline and Contributions.** We start by introducing the setting of the GKBs, the assumptions, and the learning problem (Section 3). Then, we design `GKB-UCB`, an optimistic regret minimization algorithm (Section 4) and we introduced some preliminary results (Section 5). The key contributions of this work are contained in Sections 6 and 7. In Section 6, we discuss more formally the limitations of the existing inequalities and derive a novel self-normalized Bernstein-like dimension-free inequality via the application of Freedman's inequality together with a stitching argument. In Section 7, we analyze the `GKB-UCB` with a confidence set defined in terms of the previously derived inequality and show that it achieves regret of order $\widetilde{O}(\gamma_T \sqrt{T/\kappa_*})$, being $T$ the learning horizon, $\gamma_T$ the maximal information gain, and $\kappa_*$ a term characterizing the slope of the inverse link function in the optimal decision (an efficient implementation is reported in Appendix A). This result matches the state-of-the-art of both GLBs and KBs up to multiplicative constants and logarithmic terms.

## 2   Preliminaries

**Notation.** Let $a, b \in \mathbb{N}$ with $a \le b$, we denote with $[\![a, b]\!] := \{a, a+1, \ldots, b\}$ and with $[\![b]\!] := [\![1, b]\!]$. Let $d \in \mathbb{N}$, $\mathbf{I}_d$ denotes the identity matrix of order $d$ and $\mathbf{0}_d$ the column vector of all zeros of size $d$ ($d$ is omitted when clear from the context). $\mathcal{N}(\boldsymbol{\mu}, \boldsymbol{\Sigma})$ denotes the multi-variate Gaussian distribution.

**Reproducing Kernel Hilbert Space.** Let $\mathcal{X} \subseteq \mathbb{R}^d$ be a decision set and $\mathcal{H}$ be a Hilbert space endowed with the inner product $\langle \cdot, \cdot \rangle$ (and induced norm $\| \cdot \|$). $\mathcal{H}$ is a *reproducing kernel Hilbert space* [28] if there exists a function $k : \mathcal{X} \times \mathcal{X} \to \mathbb{R}$, called *kernel*, such that it satisfies the *reproducing property*, i.e., for every function $f \in \mathcal{H}$ it holds that $f(\mathbf{x}) = \langle f, k(\mathbf{x}, \cdot) \rangle$ for every $\mathbf{x} \in \mathcal{X}$. It follows that the kernel $k$ is symmetric and satisfies the conditions for positive semi-definiteness. We denote with $I$ the identity operator on $\mathcal{H}$. From Mercer's theorem [20, 14], there exists a (possibly infinite-

---

[1]We refer to Table 1 for an overview of the properties of concentration inequalities present in the literature.

| Self-normalized Concentrations | Properties | | | | |
|---|---|---|---|---|---|
| | Condition | Dim-free | Empirical | Heterosc. | Technique |
| Dani et al., 2008 [6] | Hoeffding | ✗ | ✗ | ✗ | Freedman |
| Abbasi-Yadkori et al., 2011 [1] | Hoeffding | ✓ | ✓ | ✗ | Pseudo-Max |
| Chowdhury and Gopalan, 2017 [5] | Hoeffding | ✓ | ✓ | ✗ | Pseudo-Max |
| Faury et al., 2020 [9] | Bernstein | ✗ | ✓ | ✓ | Pseudo-Max |
| Zhou et al., 2021 [36] | Bernstein | ✗ | ✗ | ✗ | Freedman |
| Ziemann, 2024 [37] | Bernstein | ✗ | ✓ | ✗ | PAC-Bayes |
| **Our work** | **Bernstein** | ✓ | ✓ | ✓ | **Freedman** |

Table 1: Summary of the properties of self-normalized concentrations.

dimensional) feature mapping $\phi : \mathcal{X} \to \mathbb{R}^{\mathbb{N}}$ such that for every function $f \in \mathcal{H}$ there exists a (possibly infinite-dimensional) vector of coefficients $\alpha \in \mathbb{R}^{\mathbb{N}}$ such that for every $\mathbf{x} \in \mathcal{X}$, we have $f(\mathbf{x}) = \sum_{i \in \mathbb{N}} \alpha_i \phi_i(\mathbf{x}) = \langle \alpha, \phi(\mathbf{x}) \rangle$, where $\alpha$ depends on $f$ but not on $\mathbf{x}$ and for every $i \in \mathbb{N}$, we have that $\phi_i : \mathcal{X} \to \mathbb{R}$ depends on $\mathbf{x}$ but not on $f$ and the series converges absolutely and uniformly for almost all $\mathbf{x}$. Moreover, for every $i, j \in \mathbb{N}$ with $i \neq j$, we have $\|\phi_i\| = \langle \phi_i, \phi_i \rangle = 1$ and $\langle \phi_i, \phi_j \rangle = 0$, i.e., $(\phi_i)_{i \in \mathbb{N}}$ forms an orthonormal basis. Thus, if $f = \langle \alpha, \phi(\mathbf{x}) \rangle$, we have $\|f\| = \|\alpha\|$. Furthermore, for every $\mathbf{x} \in \mathcal{X}$, we have that $|f(\mathbf{x})| \leq \|f\| \|k(\cdot, \mathbf{x})\| = \|f\| \sqrt{k(\mathbf{x}, \mathbf{x})}$.

**Information Gain.** Let $k$ be a kernel, let $t \in \mathbb{N}$, and let $\mathbf{x}_1, \ldots, \mathbf{x}_t \in \mathcal{X}$ be a sequence of decisions, the *information gain* $\Gamma_t$ and the *maximal information gain* $\gamma_t$ are defined, respectively as [29]: $\Gamma_t := \frac{1}{2} \log \det(\mathbf{I} + \lambda^{-1} \mathbf{K}_t)$ and $\gamma_t := \max_{\mathbf{x}_1, \ldots, \mathbf{x}_t \in \mathcal{X}} \Gamma_t$, where $\lambda > 0$ and $\mathbf{K}_t \in \mathbb{R}^{(t-1) \times (t-1)}$ is the Kernel matrix $(\mathbf{K}_t)_{i,j} = k(\mathbf{x}_i, \mathbf{x}_j)$ for $i, j \in [\![t-1]\!]$. $\Gamma_t$ is the *mutual information* between the random vectors $\mathbf{f}_t \sim \mathcal{N}(\mathbf{0}, \nu^2 \mathbf{K}_t)$ and $\mathbf{y}_t = \mathbf{f}_t + \boldsymbol{\epsilon}_t$ where $\boldsymbol{\epsilon}_t \sim \mathcal{N}(\mathbf{0}_t, v^2 \lambda \mathbf{I}_t)$, for arbitrary $v > 0$. We use the abbreviation $\mathbf{K}_t(\lambda) := \lambda \mathbf{I} + \mathbf{K}_t$, so that, $\Gamma_t := \frac{1}{2} \log \det(\lambda^{-1} \mathbf{K}_t(\lambda))$.[2]

**Covariance Operators.** Let $\mathcal{H}$ be a RKHS with kernel $k$ inducing the feature mapping $\phi$, let $t \in \mathbb{N}$ and $\mathbf{x}_1, \ldots, \mathbf{x}_t \in \mathcal{X}$ be a sequence of decisions, the *covariance operator* is defined as: $V_t(\lambda) := V_t + \lambda I = \sum_{s=1}^{t-1} \phi(\mathbf{x}_s)\phi(\mathbf{x}_s)^\top + \lambda I$. The following identity was shown in [32]:

$$\det(\lambda^{-1} V_t(\lambda)) = \det(\lambda^{-1} \mathbf{K}_t(\lambda)). \tag{1}$$

**Canonical Exponential Family Models.** Let $f : \mathcal{X} \to \mathbb{R}$, a real-valued random variable $y$ belongs to the *canonical exponential family* [EF, 3] if it has density:

$$p(y|\mathbf{x}; f) = \exp\left(\frac{yf(\mathbf{x}) - m(f(\mathbf{x}))}{\mathfrak{g}(\tau)} + h(y, \tau)\right), \tag{2}$$

where $\tau > 0$ is a temperature parameter and $\mathfrak{g}, m : \mathbb{R} \to \mathbb{R}$ and $h : \mathbb{R}^2 \to \mathbb{R}$ are suitably defined functions [17]. This EF model allows representing a variety of distributions, including Gaussian, Bernoulli, exponentials, and Poisson. Function $m$ is called *log-partition function* and fulfills the following assumptions. As customary [17, 25], $m$ is assumed to be three times differentiable and convex. We define the *inverse link function* $\mu = m'$, that, since $m$ is convex, is monotonically non-decreasing. Thus, the following hold [17]: $\mathbb{E}[y|\mathbf{x}; f] = m'(f(\mathbf{x})) = \mu(f(\mathbf{x}))$ and $\mathbb{V}\mathrm{ar}[y|\mathbf{x}; f] = \mathfrak{g}(\tau)\dot{\mu}(f(\mathbf{x}))$. When $f$ is a linear function, the model in Equation (2) is also called *generalized linear model* [GLM, 19]. We also define the maximum slope of $\mu$, i.e., $R_{\dot{\mu}} := \sup_{f \in \mathcal{H}, \mathbf{x} \in \mathcal{X}} \dot{\mu}(f(\mathbf{x}))$.

## 3 Problem Formulation

We define the novel *generalized kernelized bandit* (GKB) setting and the learning problem.

**Setting.** Let $f^* \in \mathcal{H}$ be an unknown function belonging to the RKHS $\mathcal{H}$. At every round $t \in [\![T]\!]$, being $T \in \mathbb{N}$ the learning horizon, the learner chooses a decision $\mathbf{x}_t \in \mathcal{X}$ by means of a policy $\pi_t : \mathcal{F}_{t-1} \to \mathcal{X}$, being $\mathcal{F}_{t-1} = \sigma(\mathbf{x}_1, y_1, \ldots, \mathbf{x}_{t-1}, y_{t-1})$ the filtration of all random variables realized so far, and observes a reward $y_t \sim p(\cdot|\mathbf{x}_t; f^*)$. The goal of the agent is to find a decision $\mathbf{x}^* \in \mathcal{X}$ maximizing the expected reward: $\mathbf{x}^* \in \arg\max_{\mathbf{x} \in \mathcal{X}} \mu(f^*(\mathbf{x}))$. Since $\mu$ is monotonically non-decreasing, maximizing $\mu(f^*(\cdot))$ is equivalent to maximizing $f^*(\cdot)$. It is worth noting that the GKB generalizes two well-known settings: (i) *generalized linear bandits* [GLBs, 18] when the kernel

---

[2]Known bounds for $\gamma_t$ are available for commonly used kernels [29, 31].

123 is linear $k(\mathbf{x}, \mathbf{x}') = \langle \mathbf{x}, \mathbf{x}' \rangle$ and (ii) *kernelized bandits* [KBs, 5] when the inverse link function is the
124 identity function, i.e., $\mu = I$.

125 **Learning Problem.** We evaluate the performance of a learner, i.e., a $\pi = (\pi_t)_{t \in [\![T]\!]}$, with *cumulative*
126 *regret*: $R(\pi, T) := \sum_{t \in [\![T]\!]} (\mu(f^*(\mathbf{x}^*)) - \mu(f^*(\mathbf{x}_t)))$, where $\mathbf{x}_t = \pi_t(\mathcal{F}_{t-1})$ for all $t \in [\![T]\!]$.

127 **Assumptions.** We make the following assumptions about function $f^*$ and the RKHS $\mathcal{H}$.

128 **Assumption 3.1** (Bounded Norm). *It exists a known constant $B < +\infty$ such that $\|f^*\| \leq B$.*

129 **Assumption 3.2** (Bounded Kernel). *It exists a known constant $K < +\infty$ such that $\sup_{\mathbf{x} \in \mathcal{X}} k(\mathbf{x}, \mathbf{x}) \leq K^2$.*

130 Assumptions 3.1 and 3.2 are widely employed in the KB literature [5], where, in particular, Assump-
131 tion 3.2 is enforced with $K = 1$ and it is fulfilled by commonly used kernels (e.g., Gaussian and
132 Matérn kernels). Assumptions 3.1 and 3.2 are the analogous in GLBs of requiring the boundedness
133 of the parameter vector (since if $f = \langle \alpha, \phi \rangle$, then, $\|f\| = \|\alpha\|$) and requiring the boundedness of
134 the norm of the decisions (since when $k(\mathbf{x}, \mathbf{x}') = \langle \mathbf{x}, \mathbf{x}' \rangle$ we have that $k(\mathbf{x}, \mathbf{x}) = \|\mathbf{x}\|^2$), respec-
135 tively [2, 17]. The combination of the two allows bounding the $L_\infty$-norm of $f^*$ as $\|f^*\|_\infty \leq BK$.

136 Concerning the EF noise model, we make the following assumptions.

137 **Assumption 3.3** (Bounded noise). *Let $\mathbf{x} \in \mathcal{X}$, $y \sim p(\cdot|\mathbf{x}; f^*)$, let $\epsilon = y - \mu(f^*(\mathbf{x}))$. There exists a*
138 *known constant $R < +\infty$ such that $|\epsilon| \leq R$ almost surely.*

139 This assumption is widely used in the GLB literature [2, 25]. If we deal with $\nu^2$-subgaussian noise
140 (instead of bounded), we can take $R = \nu\sqrt{2\log(2T/\delta)}$ to ensure that $|\epsilon_t| \leq R$ uniformly for $t \in [\![T]\!]$
141 w.p. $1 - \delta$.[3] Finally, we introduce the *generalized self-concordance* property [24].

142 **Assumption 3.4** ((Generalized) Self-concordance). *There exists a known constant $R_s < +\infty$ such*
143 *that for every function $f \in \mathcal{H}$ and decision $\mathbf{x} \in \mathcal{X}$, it holds that $|\ddot{\mu}(f(\mathbf{x}))| \leq R_s \dot{\mu}(f(\mathbf{x}))$.*

144 In [25], the authors show (Lemma 2.1) that if the EF model generates random variables that are
145 bounded by $|y| \leq Y$ a.s., Assumption 3.4 hold with $R_s = Y$. Moreover, it holds for Bernoulli noise
146 with $R_s = 1$ and Gaussian with $R_s = 0$ [17].

147 **Problem Characterization.** We define the following characterizing the difficulty of the problem:
148 $\kappa_* = \frac{1}{\dot{\mu}(f^*(\mathbf{x}^*))}$ and $\kappa_\mathcal{X} = \sup_{\mathbf{x} \in \mathcal{X}} \frac{1}{\dot{\mu}(f^*(\mathbf{x}))}$. We have that $\kappa_* \leq \kappa_\mathcal{X}$. Our goal is to devise algorithms
149 for which the dominating term in the regret bound depends on $\kappa_*$ only.

## 4 Algorithm

151 In this section, we introduce Generalized
152 Kernelized Bandits-Upper Confidence
153 Bounds (GKB-UCB), a regret minimization
154 optimistic algorithm for the GKB setting
155 (Algorithm 1). GKB-UCB is composed of two
156 steps: *maximum likelihood* (ML) estimation
157 and *optimistic decision selection*. We pro-
158 vide a computationally tractable version in
159 Appendix A.

> **Input:** Decision set $\mathcal{X}$, confidence sets $\mathcal{C}_t(\delta)$
> **for** $t \in [\![T]\!]$ **do**
>     //Maximum Likelihood Estimate
>     $\widehat{f}_t \in \arg\min_{f \in \mathcal{H}} \mathcal{L}_t(f)$ (Equation 3)
>     //Optimistic Decision Selection
>     $(\widetilde{f}_t, \mathbf{x}_t) \in \arg\max_{f \in \mathcal{C}_t(\delta), \mathbf{x} \in \mathcal{X}} \mu(f(\mathbf{x}))$ (Equation 6)
>     Play $\mathbf{x}_t$ and observe $y_t$
> **end**

**Algorithm 1:** GKB-UCB.

160 **Maximum Likelihood Estimate.** At each
161 round $t \in [\![T]\!]$, we employ the samples collected
162 so far $\{(\mathbf{x}_s, y_s)\}_{s \in [\![t-1]\!]}$, to obtain an estimate $\widehat{f}_t$ of $f^*$. Starting from the EF model, we minimize
163 the *Ridge-regularized log-likelihood*:

$$\mathcal{L}_t(f) := \sum_{s=1}^{t-1} \frac{-y_s f(\mathbf{x}_s) + m(f(\mathbf{x}_s))}{\mathfrak{g}(\tau)} + \frac{\lambda}{2}\|f\|^2, \quad \forall f \in \mathcal{H}, t \in [\![T]\!], \tag{3}$$

164 where $\lambda \geq 0$ is the Ridge regularization parameter. The ML estimate is denoted as $\widehat{f}_t \in$
165 $\arg\min_{f \in \mathcal{H}} \mathcal{L}_t(f)$. Since, for Mercer's theorem, when $f \in \mathcal{H}$, we can write $f = \langle \alpha, \phi \rangle$ with

---

[3]This will result in an additional logarithmic term in the final regret bound only.

a fixed feature function $\phi$, with little abuse of notation, we can look at $\mathcal{L}_t$ as a function of the parameters $\alpha$, i.e., $\mathcal{L}_t(\alpha) \equiv \mathcal{L}_t(f)$. With this in mind, we introduce the operator $g_t(f) \in \mathbb{R}^{\mathbb{N}}$ related to the gradient of the loss $\mathcal{L}_t(f)$ w.r.t. the parameters $\alpha$ and the *weighted covariance operator* $\widetilde{V}_t(\lambda; f) \in \mathbb{R}^{\mathbb{N} \times \mathbb{N}}$ corresponding to the Hessian of the loss $\mathcal{L}_t(f)$ w.r.t. parameters $\alpha$:

$$g_t(f) := \sum_{s=1}^{t-1} \frac{\mu(f(\mathbf{x}_s))}{\mathfrak{g}(\tau)} \phi(\mathbf{x}_s) + \lambda\alpha, \qquad \nabla\mathcal{L}_t(f) = -\sum_{s=1}^{t-1} \frac{y_s\phi(\mathbf{x}_s)}{\mathfrak{g}(\tau)} + g_t(f), \tag{4}$$

$$\widetilde{V}_t(\lambda; f) := \nabla^2\mathcal{L}_t(f) = \widetilde{V}_t(f) + \lambda I = \sum_{s=1}^{t-1} \frac{\dot{\mu}(f(\mathbf{x}_s))}{\mathfrak{g}(\tau)} \phi(\mathbf{x}_s)\phi(\mathbf{x}_s)^\top + \lambda I. \tag{5}$$

The loss function $\mathcal{L}_t$ and the operators $g_t$ and $\widetilde{V}_t$ defined above reduce to the ones employed for GLBs under the assumption that the kernel $k$ is the linear one [2, 9, 17]. Furthermore, if $\mu = I$, we have that $\widetilde{V}_t(\lambda; f) = V_t(\lambda)$, i.e., the covariance operator.

**Optimistic Decision Selection.** Once the ML function $\widehat{f}_t$ is computed, the algorithm chooses an optimistic function $\widetilde{f}_t \in \mathcal{H}$ in a suitable confidence set $\mathcal{C}_t(\delta)$, together with the optimistic choice $\mathbf{x}_t$:

$$(\widetilde{f}_t, \mathbf{x}_t) \in \underset{f\in\mathcal{C}_t(\delta),\, \mathbf{x}\in\mathcal{X}}{\arg\max} \mu(f(\mathbf{x})). \tag{6}$$

It is worth noting that since $\mu$ is non-decreasing, we can ignore $\mu$ in the maximization. We will consider a confidence set, defined for every round $t \in [\![T]\!]$ and confidence $\delta \in (0,1)$ as follows:[4]

$$\mathcal{C}_t(\delta) = \left\{ f \in \mathcal{H} \,:\, \left\| g_t(f) - g_t(\widehat{f}_t) \right\|_{\widetilde{V}_t^{-1}(\lambda;f)} \le B_t(\delta; f) \right\}, \tag{7}$$

where the confidence ratio $B_t(\delta; f)$ will be specified later with the goal of guaranteeing optimism, i.e., that the true unknown function $f^*$ belongs to $\mathcal{C}_t(\delta)$ in high probability, and limiting the regret.

## 5 Weighted Kernel

We discuss how the combination between a function $f \in \mathcal{H}$ with an inverse link function $\mu$ induced another RKHS space that can be characterized by its *weighted kernel*. Let $f \in \mathcal{H}$, we define the weighted feature mapping (now dependent on $f$) for every $\mathbf{x} \in \mathcal{X}$ as: $\widetilde{\phi}(\mathbf{x}; f) = \sqrt{\dot{\mu}(f(\mathbf{x}))\mathfrak{g}(\tau)^{-1}}\phi(\mathbf{x})$. In the primal (feature) space, this allows looking at the weighted covariance operator $\widetilde{V}_t(\lambda; f)$ as the covariance operator induced by the feature mapping $\widetilde{\phi}(\cdot; f)$, i.e., $\widetilde{V}_t(\lambda; f) = \sum_{s=1}^{t-1} \widetilde{\phi}(\mathbf{x}_s; f)\widetilde{\phi}(\mathbf{x}_s; f)^\top + \lambda I$. Passing to the dual (kernel) space, we define the weighted kernel as:

$$\widetilde{k}(\mathbf{x}, \mathbf{x}'; f) := \langle \widetilde{\phi}(\mathbf{x}; f), \widetilde{\phi}(\mathbf{x}'; f) \rangle = \mathfrak{g}(\tau)^{-1} \sqrt{\dot{\mu}(f(\mathbf{x}))} k(\mathbf{x}, \mathbf{x}') \sqrt{\dot{\mu}(f(\mathbf{x}'))}, \quad \forall \mathbf{x}, \mathbf{x}' \in \mathcal{X}. \tag{8}$$

This is, in all regards, a valid kernel since it is obtained starting from a valid kernel and performing a legal transformation [28]. This way, we can define the weighted kernel matrix as $\widetilde{\mathbf{K}}_t(\lambda; f) = \lambda\mathbf{I} + \widetilde{\mathbf{K}}_t(f)$, where $\widetilde{\mathbf{K}}_t(f) = (\widetilde{k}(\mathbf{x}_i, \mathbf{x}_j; f))_{i,j\in[\![t-1]\!]}$. Using the identity in Equation (1), we can also deduce that $\det(\lambda^{-1}\widetilde{V}_t(\lambda; f)) = \det(\lambda^{-1}\widetilde{\mathbf{K}}_t(\lambda; f))$. We also define the *weighted information gain* $\widetilde{\Gamma}_t(f)$ and the *weighted maximal information gain* $\widetilde{\gamma}_t(f)$ as $\widetilde{\Gamma}_t(f) := \frac{1}{2}\log\det(\lambda^{-1}\widetilde{\mathbf{K}}_t(\lambda; f))$ and $\widetilde{\gamma}_t(t) := \max_{\mathbf{x}_1,\ldots,\mathbf{x}_t\in\mathcal{X}} \widetilde{\Gamma}_t(f)$. Finally, we consider the maximum value of the (maximal) information gain by varying the function $f$ in $\mathcal{H}$, i.e., $\widetilde{\Gamma}_t(\mathcal{H}) = \sup_{f\in\mathcal{H}} \widetilde{\Gamma}_t(f)$ and $\widetilde{\gamma}_t(\mathcal{H}) = \sup_{f\in\mathcal{H}} \widetilde{\gamma}_t(f)$. The following result relates weighted and unweighted information gains.

**Lemma 5.1.** *Let $\mathcal{H}$ be a RKHS induced by kernel $k$. Let $t \in \mathbb{N}$ and let $\mathbf{x}_1,\ldots,\mathbf{x}_t \in \mathcal{X}$ be a sequence of decisions. It holds that $\widetilde{\Gamma}_t(\mathcal{H}) \le \max\{1, R_{\dot{\mu}}\mathfrak{g}(\tau)^{-1}\}\Gamma_t$.*

Notice that the bound introduces just a dependence on the maximum slope of the inverse link function $R_{\dot{\mu}}$ and no dependence on the minimum slope $\kappa_{\mathcal{X}}$. This result will play a significant role in the derivation of an efficient implementation for GKB-UCB (Appendix A).

---

[4]Assessing whether a function $f \in \mathcal{H}$ belongs to the confidence set $\mathcal{C}_t(\delta)$ is clearly intractable since it requires computing norms of operators. In Appendix A, we provide an efficient alternative confidence set that will lead to analogous regret guarantees.

# 6 Challenges and New Technical Tools

In this section, we discuss the main challenges for achieving sensible regret guarantees for GKBs. We start discussing the limitations of existing *self-normalized* concentration bounds (see Table 1) to control the error in the ML estimate (Section 6.1). This motivates the need for a novel self-normalized inequality that represents a key contribution of this work (Section 6.2).

## 6.1 Limitations of Existing Self-Normalized Concentration Inequalities

To understand the need for a novel concentration bound, we need to anticipate some key passages of the regret analysis. We recall that the confidence radius $B_t(\delta; f)$ should be designed to guarantee that: $(i)$ the true unknown function $f^* = \langle \alpha^*, \phi \rangle$ belongs to $\mathcal{C}_t(\delta)$ (Equation 7) and $(ii)$ the regret is as small as possible. For point $(i)$, we can conveniently express the difference between the operators $g_t$ evaluated in the true function $f^*$ and in the ML estimate $\widehat{f}_t$ (see Lemma 7.1):

$$g_t(f^*) - g_t(\widehat{f}_t) = \mathfrak{g}(\tau)^{-1} \sum_{s=1}^{t-1} \epsilon_s \phi(\mathbf{x}_s) + \lambda \alpha^*, \qquad (9)$$

where $\epsilon_s = y_s - \mu(f^*(\mathbf{x}_s))$ is the noise. Thus, since since $\alpha^*$ is bounded in norm under Assumption 3.1, to suitably design $B_t(\delta; f)$, we need to control the martingale $S_t = \sum_{s=1}^{t-1} \epsilon_s \phi(\mathbf{x}_s)$. For point $(ii)$, in the regret analysis, we need to bound the difference between optimistic function $\widetilde{f}_t$ and true unknown function $f^*$, both evaluated in the played decision $\mathbf{x}_t$, i.e., $\widetilde{f}_t(\mathbf{x}_t) - f^*(\mathbf{x}_t)$ with the martingale $S_t$. Similarly to [2, 9], this is done by decomposing both functions as an inner product (Mercer's theorem) and then applying a Cauchy-Schwarz inequality by making a *specific* choice of operator $W_t(f^*)$, possibly depending on the unknown function $f^*$:

$$\widetilde{f}_t(\mathbf{x}_t) - f^*(\mathbf{x}_t) = \langle \widetilde{\alpha}_t - \alpha^*, \phi(\mathbf{x}_t) \rangle \leq \underbrace{\|\widetilde{\alpha}_t - \alpha^*\|_{W_t(f^*)}}_{(A)} \underbrace{\|\phi(\mathbf{x}_t)\|_{W_t(f^*)^{-1}}}_{(B)}. \qquad (10)$$

The choice of operator $W_t(f^*)$ has two effects: $(i)$ by relating term (A) with the confidence set $\mathcal{C}_t(\delta)$ definition, it determines the multiplicative coefficient and the norm under which martingale $S_t$ has to be controlled and $(ii)$ it allows bounding (B) by means of an *elliptic potential lemma* [16, Lemma 19.4]. We now discuss two choices of operators $W_t(f^*)$ leading to different concentration bounds and, consequently, confidence sets, and discuss their advantages and disadvantages.

**Covariance Operator** $(W_t(f^*) = V_t(\lambda))$**.** We start considering the case in which $W_t(f^*) = V_t(\lambda)$, where $V_t$ is the usual covariance operator. In this case, we can link the term (A) with the confidence set as follows (see Lemma C.4):

$$(A) = \|\widetilde{\alpha}_t - \alpha^*\|_{V_t(\lambda)} \leq (1 + 2R_s BK) \max\{1, \mathfrak{g}(\tau)\textcolor{red}{\kappa_{\mathcal{X}}}\} \left\| g_t(\widetilde{f}_t) - g_t(f^*) \right\|_{V_t^{-1}(\lambda)}, \qquad (11)$$

introducing an inconvenient multiplicative dependence on $\max\{1, \mathfrak{g}(\tau)\kappa_{\mathcal{X}}\}$, i.e., on the minimum slope $\kappa_{\mathcal{X}}$ of the inverse link function. At this point, we have to control the martingale $S_t$ under the norm weighted by $V_t^{-1}(\lambda)$, as $\left\| g_t(f^*) - g_t(\widehat{f}_t) \right\|_{V_t^{-1}(\lambda)} \leq \|S_t\|_{V_t^{-1}(\lambda)} + \frac{B}{\sqrt{\lambda}}$. The quantity $\|S_t\|_{V_t^{-1}(\lambda)}$ can be conveniently bounded by using a self-normalized concentration bound for sub-gaussian[5] martingales (i.e., *Hoeffding-like*), as in the seminal work [1]:

$$\|S_t\|_{V_t^{-1}} \leq R\sqrt{2\log(\delta^{-1}) + \log\det(\lambda^{-1}V_t(\lambda))} = R\sqrt{2\log(\delta^{-1}) + \log\det(\lambda^{-1}\mathbf{K}_t(\lambda))}, \quad (12)$$

where the equality is obtained by Equation (1). We recall that the second bound is also obtained in Theorem 1 of [5] where the quantity $\|S_t\|_{V_t^{-1}}$ is controlled in the dual (kernel) space. The advantage of these bounds is that they do not exhibit a dependence on the dimensionality $d$ of the feature space $\phi$, which in GKBs is infinite. Nevertheless, in this way, the dependence on the minimum slope of the inverse link function $\kappa_{\mathcal{X}}$ (as in Equation 11) becomes unavoidable in the regret. This suggests that we should prefer a different choice of operator $W_t(f^*)$.

**Weighted Covariance Operator** $(W_t(f^*) = \widetilde{V}_t(\lambda; f^*))$**.** The presence of the multiplicative factor $\kappa_{\mathcal{X}}$ depends on the covariance operator and emerges also in the GLB setting when making the choice

---

[5]We recall that since $|\epsilon_s| \leq R$ a.s., it is also $R^2$-subgaussian.

$W_t(f^*) = V_t(\lambda)$ [2, 9]. The solution, in the GLB case, consists of choosing the weighted covariance operator $W_t(f^*) = \widetilde{V}_t(\lambda; f^*)$, where each outer product $\phi(\mathbf{x}_s)\phi(\mathbf{x}_s)^\top$ is weighted by the variance $\frac{\dot{\mu}(f(\mathbf{x}_s))}{\mathfrak{g}(\tau)}$ of the noise random variable $\epsilon_s$. This allows relating the distance of the parameters with the confidence set $\mathcal{C}_t(\delta)$, avoiding the inconvenient dependence on $\kappa_{\mathcal{X}}$ (see Lemma C.4 with $f'' = f$):

$$(\text{A}) = \|\widetilde{\alpha}_t - \alpha^*\|_{\widetilde{V}_t(\lambda; f^*)} \leq (1 + 2R_s BK) \left\| g_t(\widehat{f}_t) - g_t(f^*) \right\|_{\widetilde{V}_t^{-1}(\lambda; f^*)}. \tag{13}$$

Proceeding analogously as above, we should now control the quantity $\|S_t\|_{\widetilde{V}_t^{-1}(\lambda; f^*)}$. Since the weighted covariance operator $\widetilde{V}_t(\lambda; f^*)$ contains the variance of each sample, we need to resort to a *Bernstein-like* self-normalized concentration bound in order to make effective use of such information. The fundamental result in the GLB literature is the bound of [9, Theorem 1]:

$$\|S_t\|_{\widetilde{V}_t^{-1}(\lambda; f^*)} \leq \frac{\sqrt{\lambda}}{2} + \frac{2}{\sqrt{\lambda}}\boldsymbol{d}\log 2 + \frac{2}{\sqrt{\lambda}}\log\frac{1}{\delta} + \frac{1}{\sqrt{\lambda}}\log\det(\lambda^{-1}\widetilde{V}_t(\lambda; f^*)), \tag{14}$$

where $d$ is the dimensionality of the feature map $\phi$, which is infinite-dimensional in our GKB setting, making the bound vacuous.[6]

## 6.2 A Novel Bernstein-like Dimension-Free Self-Normalized Inequality

From the above discussion, it should now appear clear why we need a *novel self-normalized concentration bound* that combines two desired properties:

- *Bernstein-like*: it should account for a weighted covariance operator $\widetilde{V}_t(\lambda; f^*)$ where the weights correspond to the variance of the samples to avoid the inconvenient multiplicative factor $\kappa_{\mathcal{X}}$;
- *Dimension-free*: it should avoid any dependence on the dimensionality of the feature space $\phi$, in order to make it applicable to our GKB setting, where $\phi$ can be infinite-dimensional.

With this goal, we deviate from the two traditional approaches to derive self-normalized concentrations, i.e., *pseudomaximization* via method of mixtures [1, 7, 9] and *PAC Bayes* [17, 37]. Instead, we follow the path of [36] that, in turn, extends [6], by directly decomposing the norm $\|S_t\|_{\widetilde{V}_t^{-1}(\lambda; f^*)}$ and bounding individual terms by means of Freedman's inequality [11]. In addition to the requirements above, we aim to obtain a *data-driven* bound in which, just like in Equations (12) and (14), the bound depends on the sequence of the actual decisions, i.e., on the weighted information gain $\widetilde{\Gamma}_t(f^*) = \frac{1}{2}\log\det(\lambda^{-1}\widetilde{V}_t(\lambda; f^*))$ instead of the *maximal* information gain $\widetilde{\gamma}_t(f^*)$. This is clearly desirable since $\widetilde{\Gamma}_t(f^*) \leq \widetilde{\gamma}_t(f^*)$.[7] However, this is not straightforward when following the technique of [6, 36], that necessitates deterministic bounds to the cumulative variance for the application of Freedman's inequality. For this reason, we provide a first result that extends Freedman's inequality allowing for bounds of the cumulative variance that are not deterministic but, instead, predictable processes. This will represent the core for deriving our self-normalized concentration bound.

**Theorem 6.1** (A data-driven Freedman's inequality). *Let $(z_t)_{t\geq 1}$ be a real-valued martingale difference sequence adapted to the filtration $\mathcal{F}_t$ such that $z_t \leq R$ a.s. for all $t \geq 1$. Let $(v_t)_{t\geq 1}$ be a process predictable by the filtration $\mathcal{F}_t$ such that for every $t \geq 1$, we have that $\sum_{s=1}^{t} \mathbb{E}[z_s^2|\mathcal{F}_{s-1}] \leq v_t$ a.s.. Then, for every $\eta > 1$ and $v_0 > 0$, with probability at least $1 - \delta$, it holds that:*

$$\forall t \geq 1: \qquad \sum_{s=1}^{t} z_s \leq \sqrt{2\max\{v_0, \eta v_t\}\log\frac{\pi^2(\widehat{\ell}+1)^2}{6\delta}} + \frac{R}{3}\log\frac{\pi^2(\widehat{\ell}+1)^2}{6\delta}, \tag{15}$$

*where $\widehat{\ell} = \max\left\{0, \lceil\log_\eta(v_t/v_0)\rceil\right\}$.*

The inequality of Theorem 6.1, compared to the standard Freedman's inequality (see Lemma B.1), allows obtaining a bound that depends on the predictable process $v_t$ that we can think to as a proxy (upper bound) of the variance that, however, does not need to be deterministic. This allows us to obtain bounds that depend on the actual sequence of decisions $\mathbf{x}_1, \ldots, \mathbf{x}_t$ and their weighted information

---

[6]One could attempt to operate as in [9, Theorem 1] for deriving but directly in the dual (kernel) space. Although this is possible, it would make appear a dependence on the order of the weighted kernel matrix $\widetilde{\mathbf{K}}_t(\lambda; f)$, i.e., $t$ in replacement of $d$. This is not of any help since it will make the regret degenerate to linear.

[7]Indeed, in [36], the bound depends on an upper bound of $\gamma_t$ obtained by bounding the maximum value of $\log\det(\lambda^{-1}V_t)$ considering the worst-case sequence of decisions [see Lemma B.2 of 36].

gain $\widetilde{\Gamma}_t(f^*)$ rather than on the maximal weighted information gain $\widetilde{\gamma}_t(f^*)$, with an improvement over previous inequalities like [36]. From a technical perspective, Theorem 6.1 is obtained using a *stitching* argument [13] that brings two beneficial effects. First, it allows to accurately perform *union bounds* considering the values that the predictable process can take over a geometric grid $\{\eta^\ell v_0 \: : \: \ell \in \mathbb{N}\}$ enabling the use of the data-driven quantity $v_t$, where the parameters $\eta > 1$ and $v_0 > 0$ can be selected to tighten the bound. Second, it allows replacing a $\log t$ term in the bound with a $\log \log t$ at the price of a larger multiplicative constant $\eta > 1$. A similar data-driven result has been provided in [12, Theorem 12]. However, our result allows tuning the parameters $\eta$ and $v_0$ to tighten the bound, ultimately leading to an improvement of the constants. We can now use Theorem 6.1 to derive our novel *self-normalized Bernstein-like dimension-free* concentration inequality.

**Theorem 6.2** (Bernstein-Like Dimension-Free Self-Normalized Concentration). *Let $(\mathbf{x}_t)_{t\geq 1}$ be a discrete-time stochastic process predictable by the filtration $\mathcal{F}_t$ and let $(\epsilon_t)_{t\geq 1}$ be a real-valued stochastic process adapted to the $\mathcal{F}_t$ such that $\mathbb{E}[\epsilon_t|\mathcal{F}_{t-1}] = 0$, $\mathbb{V}\mathrm{ar}[\epsilon_t|\mathcal{F}_{t-1}] = \sigma_t^2 = \sigma^2(\mathbf{x}_t)$, and $|\epsilon_t| \leq R$ a.s. for every $t \geq 1$. Let $\phi : \mathcal{X} \to \mathbb{R}^\mathbb{N}$ be the feature mapping induced by kernel $k$ such that $\|\phi(\mathbf{x})\|_2 \leq K$ for every $\mathbf{x} \in \mathcal{X}$. Let:*

$$S_t := \sum_{s=1}^{t-1} \epsilon_s \phi(\mathbf{x}_s), \qquad \widetilde{V}_t(\lambda) := \sum_{s=1}^{t-1} \sigma_s^2 \phi(\mathbf{x}_s)\phi(\mathbf{x}_s)^\top + \lambda I. \tag{16}$$

*Then, for every $\delta \in (0,1)$ and $t \geq 1$, with probability at least $1 - \delta$ it holds that:*

$$\|S_t\|_{\widetilde{V}_t^{-1}(\lambda)} \leq \left( \sqrt{73 \log \det(\lambda^{-1}\widetilde{V}_t)} + \sqrt{3} \right) \sqrt{\log \frac{\pi^2(\rho+1)^2}{3\delta}} + \frac{3RK}{\sqrt{\lambda}} \log \frac{\pi^2(\rho+1)^2}{3\delta}, \tag{17}$$

*where $\rho = \max\left\{ 0, \left\lceil \log\left( \frac{8R^2K^2(t-1)^3}{\lambda} \log\left(1 + \frac{K^2R^2}{\lambda}\right) \right) \right\rceil \right\}$.*

The concentration bound, as desired, displays no dependence on the dimensionality $d$ of the feature map $\phi$ and no explicit dependence on $t$ (apart from sub-logarithmic ones). We succeeded to remove the dependence from $d$ by replacing it with the norm of the feature map, which is bounded by $K$ under Assumption 3.1. It is worth noting that, thanks to the data-driven bound of Theorem 6.1, we have a dependence on the term $\log \det(\lambda^{-1}\widetilde{V}_t(\lambda))$ that, thanks to the identity in Equation (1), can be expressed in the dual (kernel) space by means of the information gain $2\widetilde{\Gamma}_t = \log \det(\lambda^{-1}\widetilde{\mathbf{K}}_t(\lambda))$, where the weighted kernel matrix $\widetilde{\mathbf{K}}_t(\lambda)$ is obtained by means of the weighted kernel $\widetilde{k}(\mathbf{x}, \mathbf{x}') = \sigma(\mathbf{x})k(\mathbf{x}, \mathbf{x}')\sigma(\mathbf{x}')$ that induces the modified feature map $\widetilde{\phi}(\mathbf{x}) = \sigma(\mathbf{x})\phi(\mathbf{x})$. By denoting with $\widetilde{\gamma}_t = \max_{\mathbf{x}_1,\ldots,\mathbf{x}_t \in \mathcal{X}} \widetilde{\Gamma}_t$, we can write the non-data-driven bound, holding with probability $1 - \delta$:

$$\forall t \geq 1 : \qquad \|S_t\|_{\widetilde{V}_t^{-1}} \leq \left( \sqrt{146\widetilde{\gamma}_t} + \sqrt{3} \right) \sqrt{\log \frac{\pi^2(\rho+1)^2}{3\delta}} + \frac{3RK}{\sqrt{\lambda}} \log \frac{\pi^2(\rho+1)^2}{3\delta}. \tag{18}$$

## 7 Regret Analysis

In this section, we provide the regret analysis of GKB-UCB (Algorithm 1). We start with a lemma to show that $f^*$ belongs to the confidence set $\mathcal{C}_t(\delta)$ (in high probability) with a proper choice of the confidence radius $B_t(\delta; f)$ (Lemma 7.1). Then, we move to the regret analysis (Theorem 7.2).

**Lemma 7.1** (Good Event). *Let $t \in \mathbb{N}$, $f \in \mathcal{H}$, and $\delta \in (0,1)$, define the confidence radius as:*

$$B_t(\delta; f) := \sqrt{\lambda}B + \frac{1}{\mathfrak{g}(\tau)}\left( \sqrt{73 \log \det(\lambda^{-1}\widetilde{V}_t(\lambda; f))} + \sqrt{3} \right)\sqrt{\log \frac{\pi^2(\rho+1)^2}{3\delta}} + \frac{3RK}{\mathfrak{g}(\tau)\sqrt{\lambda}} \log \frac{\pi^2(\rho+1)^2}{3\delta},$$

*where $\rho = \max\left\{ 0, \left\lceil \log\left( \frac{8R^2K^2(t-1)^3}{\lambda} \log\left(1 + \frac{K^2R^2}{\lambda}\right) \right) \right\rceil \right\}$. Let $\mathcal{E}_\delta := \{\forall t \geq 1 \: : \: f^* \in \mathcal{C}_t(\delta)\}$. Under Assumptions 3.1, 3.2, and 3.3, it holds that $\Pr(\mathcal{E}_\delta) \geq 1 - \delta$.*

Lemma 7.1 resorts to our novel self-normalized bound (Theorem 6.2), together with Assumption 3.1, to provide a form to the confidence radius $B_t(\delta; f)$. It is worth noting that, differently from the majority of existing works [1, 2, 17], $B_t(\delta; f)$ explicitly depends on function $f$ since the operator $\widetilde{V}_t(\lambda; f)$ necessitates $f$ to compute the weights $\mathfrak{g}(\tau)^{-1}\dot{\mu}(f(\mathbf{x}_s))$. By exploiting the identity in Equation (1), we can move to the dual (kernel) space in order to operate with finite-dimensional objects: $\log \det(\lambda^{-1}\widetilde{V}_t(\lambda; f)) = \log \det(\widetilde{\mathbf{K}}_t(\lambda; f)) = 2\widetilde{\Gamma}_t(f)$. Let us also define its worst-case

version w.r.t. the choice of function $f \in \mathcal{H}$, i.e., $B_t(\delta; \mathcal{H}) = \sup_{f \in \mathcal{H}} B_t(\delta; f)$. Although GKB-UCB makes use of the confidence radius $B_t(\delta; f)$, for analysis purposes, we also define a non-data-driven confidence radius, where the information gain $\widetilde{\Gamma}_t(f)$ is replaced by its maximal version:

$$\beta_t(\delta; f) \coloneqq \sqrt{\lambda} B + \mathfrak{g}(\tau)^{-1} \left( \sqrt{146 \widetilde{\gamma}_t(f)} + \sqrt{3} \right) \sqrt{\log \frac{\pi^2 (\rho + 1)^2}{3\delta}} + \frac{3\mathfrak{g}(\tau)^{-1} RK}{\sqrt{\lambda}} \log \frac{\pi^2 (\rho + 1)^2}{3\delta},$$

and, finally, we introduce its worst-case version w.r.t. the choice of function $f \in \mathcal{H}$, i.e., $\beta_t(\delta; \mathcal{H}) = \sup_{f \in \mathcal{H}} \beta_t(\delta; f)$, i.e., obtained from $\beta_t(\delta; f)$ by replacing $\widetilde{\gamma}_t(f)$ with $\widetilde{\gamma}_t(\mathcal{H})$.

We are now ready to present the regret bound of GKB-UCB.

**Theorem 7.2** (Regret Bound of GKB-UCB). *Under Assumptions 3.1, 3.2, 3.3, and 3.4, GKB-UCB with the confidence radius $B_t(\delta; f)$ as defined in Lemma 7.1 and $\lambda > 0$, for every $\delta \in (0, 1)$, with probability at least $1 - \delta$, suffers regret bounded as $R(\text{GKB-UCB}, T) = R_{\text{perm}}(T) + R_{\text{trans}}(T)$, where:*

$$R_{\text{perm}}(T) \le 8(1 + 2R_s BK)\beta_T(\delta; \mathcal{H}) \sqrt{\max\left\{\mathfrak{g}(\tau), \lambda^{-1} R_{\dot{\mu}} K^2\right\} \widetilde{\gamma}_T(f^*)} \sqrt{\frac{T}{\kappa_*}}, \tag{19}$$

$$R_{\text{trans}}(T) \le 32 R_s (1 + R_{\dot{\mu}} \kappa_{\mathcal{X}})(1 + 2R_s BK)^2 \beta_T(\delta; \mathcal{H})^2 \max\left\{\mathfrak{g}(\tau), \lambda^{-1} R_{\dot{\mu}} K^2\right\} \widetilde{\gamma}_T(f^*). \tag{20}$$

The proof schema of Theorem 7.2 follows similar steps to [2] and the result, indeed, displays an analogous regret decomposition into a *permanent* term $R_{\text{perm}}(T)$ and a *transient* term $R_{\text{trans}}(T)$. Regarding the dependence on explicit $T$ and $\kappa_*$, $R_{\text{perm}}(T)$ is the dominating term that displays the desired dependence on $\sqrt{T/\kappa_*}$, whereas $R_{\text{trans}}(T)$ exhibits a dependence on the minimum slope of the inverse link function $\kappa_{\mathcal{X}}$, but has only logarithmic dependence on $T$ and, for this reason, it is negligible. To highlight the dependence on the information gain, we explicit the form of the individual terms in the case $\lambda \ge \Omega(K^2)$:[8] $\beta_T(\delta; \mathcal{H}) = \widetilde{O}(\sqrt{\lambda} B + \sqrt{\widetilde{\gamma}_T(\mathcal{H}) \log(\delta^{-1})} + RK \log(\delta^{-1}))$. Thus, we obtain a regret bound of order:

$$R(\text{GKB-UCB}, T) \le \widetilde{O}\left( (1 + R_s BK)\left(\sqrt{\lambda} B + \sqrt{\widetilde{\gamma}_T(\mathcal{H}) \log(\delta^{-1})} + RK \log(\delta^{-1})\right) \sqrt{\widetilde{\gamma}_T(f^*)} \sqrt{\frac{T}{\kappa_*}} \right).$$

We have two terms related to the weighted information gain, i.e., $\widetilde{\gamma}_T(\mathcal{H})$ and $\widetilde{\gamma}_T(f^*)$. This is due to the fact that our weighted kernel $\widetilde{k}(\cdot, \cdot; f)$ explicitly depends on the evaluated function $f$. It is worth noting that, thanks to Lemma 5.1, we can bound both with the (unweighted) information gain as $\widetilde{\gamma}_T(f^*) \le \widetilde{\gamma}_T(\mathcal{H}) \le \max\{1, R_{\dot{\mu}} \mathfrak{g}(\tau)^{-1}\} \gamma_T$ at the mild price of a multiplicative term.

Let us now comment on the tightness of the bound in the particular cases of KBs and GLBs. For KBs, we are in the presence of $\nu^2$-subgaussian noise and, thus, we need to set $R = O(\nu \sqrt{\log(T/\delta)})$. Furthermore, we have that $R_s = 0$ and $\mu = I$ (consequently, $\dot{\mu} = 1$, $\kappa_* = 1$, and $\widetilde{\gamma}_T(f^*) = \widetilde{\gamma}_T(\mathcal{H}) = \gamma_T$). This allows recovering the bound of order $\widetilde{O}\left( \left( \sqrt{\lambda} B + \sqrt{\gamma_T \log(\delta^{-1})} + K\nu \log(\delta^{-1})^{3/2} \right) \sqrt{\gamma_T T} \right)$, matching the regret order of [5] up to logarithmic terms. For GLBs, we can bound the information gain as (see Lemma 11 of [2]):

$$\widetilde{\gamma}_T(\mathcal{H}) \le \max\{1, R_{\dot{\mu}} \mathfrak{g}(\tau)^{-1}\} \gamma_T \le \max\{1, R_{\dot{\mu}} \mathfrak{g}(\tau)^{-1}\} d \log\left(\lambda + \frac{TK^2}{d}\right). \tag{21}$$

This leads to bound of order $\widetilde{O}((1 + R_s BK)(\sqrt{\lambda} B + \sqrt{d \log(\delta^{-1})} + RK \log(\delta^{-1}))\sqrt{dT/\kappa_*})$, matching the result of [2] up to logarithmic terms.

## 8 Conclusions

In this paper, we have introduced the novel setting of GKBs, unifying KBs and GLBs. We have provided a novel Bernstein-like dimension-free self-normalized bound of independent interest. We employed it to analyze the regret of GKB-UCB showing tight regret bounds. Future works include investigating the use of the techniques from [17] in order to remove the multiplicative dependence on the norm and kernel bounds $(1 + R_s BK)$ in the regret bound as well as the study of the inherent complexity of regret minimization in the GLB setting by conceiving regret lower bounds [26].

---

[8]With the $O(\cdot)$ notation, we suppress multiplicative constants and dependencies on $\mathfrak{g}(\tau)$ and $R_{\dot{\mu}}$. With the $\widetilde{O}(\cdot)$ notation, we also suppress logarithmic dependencies on all variables, except for $\delta$.

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
