# OpenReview forum: "Generalized Kernelized Bandits: Self-Normalized Bernstein-Like Dimension-Free Inequality and Regret Bounds"
_NeurIPS.cc/2025/Conference — Submitted to NeurIPS 2025_

### Official Review · Reviewer_xX8K · 2025-07-02

**Clarity:** 3
**Significance:** 3
**Originality:** 2
**Rating:** 5
**Confidence:** 4

**Summary:**

The paper proposes and tackles RKHS version of generalized linear bandit (GLB), which unifies both the finite-dimensional GLB and kernelized linear bandits. The main technical novelty is the derivation of a new self-normalized, Bernstein-type concentration bound by combining the "stitching" argument of Howard et al. (2021) with Freedman's inequality. Based on this, the authors propose GKB-UCB that attains a regret bound of $\tilde{O}(\gamma_T \sqrt{T / \kappa_*})$, where $\gamma_T$ is the maximal information gain and $\kappa_*$ is the inverse of slope at the optimal arm, reminiscent and strict generalization of the best known regret for finite-dimensional GLB.

**Questions:**

1. The regret bounds, when instantiated to finite-dimensional scenarios, seem to have extra logarithmic factors. What is the reason for this? Is this an inherent limitation of the algorithm, or simply an artifact of the current proof?
2. The authors state that for the analysis purpose, the worst-case radius $B_t(\delta, \mathcal{H}) = \max_{f \in \mathcal{H}} B_t(\delta, f)$ is used. Why is this necessary? I don't recall seeing similar arguments when proving the regret bounds for finite-dimensional GLBs (please correct me if I'm wrong here)
3. The data-driven Freedman's inequality, although itself seems novel, seems to resemble prior such bounds: [1, Theorem 1], [2, Lemma 3] with simpler proofs. Can the authors elaborate on why the current form of the data-driven Freedman is necessarily tight for this paper's purpose?


[1] https://proceedings.mlr.press/v15/beygelzimer11a.html

[2] https://proceedings.mlr.press/v238/lee24d.html

**Ethical Concerns:**

["NO or VERY MINOR ethics concerns only"]

**Final Justification:**

All my concerns have been addressed, and considering the novelty of the tackled problem setting and its success, I maintain my score of 5.

There was some extensive discussion between the authors and Reviewer nkzS, which I have fully reviewed and commented on as well. While I do agree with some of the points that nkzS has raised, I do not agree with the nkzS's overall downplaying of this paper's contribution, as none of the prior works' notions of effective dimension do not consider the heteroskedasticity (caused by GLM) properly. Also, I believe that the authors' rebuttal has sufficiently clarified regarding all the confusions that nkzS has raised.

**Limitations:**

yes

**Quality:**

3

**Strengths And Weaknesses:**

**Strengths**
- Well-written
- Technically sound, and the proposed approach to building new concentration seems distinct from prior works.


**Weaknesses**
- No experimental results. Maybe take some inspirations from [1]. I feel that this will strengthen the paper, especially as it highlights efficient implementation of GKB-UCB. I would be interested to see the empirical regret comparison between the non-efficient ver and efficient ver, and other applicable baselines.


[1] https://proceedings.mlr.press/v70/chowdhury17a

---

> ### Author Rebuttal · Authors · 2025-07-31
>
> We thank the Reviewer for the feedback and for having recognized that the paper is well-written and teachnically sound. Below we address the Reviewer's concerns.
>
> ### Weaknesses
>
> Although we believe that the core contributions of the paper are mainly theoretical and algorithmic, we agree that some numerical simulations comparing the two versions of our algorithm, as well as comparisons with existing ones for KBs and GLBs, would be of interest. We are running these simulations, and we will include them in the final version of the paper.
>
> ### Questions
>
> 1. Yes, especially in the case of sub-Gaussian noise, we incur an additional $\log(\delta^{-1})$ factor (see line 340). We believe this is due to the use of Bernstein-like concentration bounds, which are suboptimal when dealing with sub-Gaussian noise. In this case, Hoeffding-like concentration bounds would be more appropriate.
> However, for our purposes, we need to use Bernstein-like bounds in order to highlight the dependence on the variance—specifically, to obtain tight bounds that depend on $\kappa_*$ in the dominating term.
>
>
> 2. This is necessary because the confidence radius $B_t(\delta; f)$ is instantiated for different functions $f$, specifically the optimistic ones $\widetilde{f}_t$ (see, for instance, Equation 105). To avoid a dependence on the sequence $(\\widetilde{f}\_t)\_{t \in [T]}$, which is random, in the regret bound, we chose to use the worst-case version with respect to the choice of $f$, i.e., $B_t(\delta, \mathcal{H})$.
> This phenomenon does not arise in generalized linear bandit algorithms, since the corresponding confidence radius does not explicitly depend on $\theta$ (the counterpart of $f$ in that setting); see, for instance, [A, Section 3.1]. In other words, those algorithms already consider the worst-case confidence radius by design.
>
>
> 3. Informally, our inequality can be seen as an extension of those mentioned by the Reviewer, allowing us to achieve a tight dependence on the square root of the cumulative variance $V_t = \sum_{s=1}^t \mathbb{E}[X_s^2 \mid \mathcal{F}_{s-1}]$. Specifically:
> - [1, Theorem 1]: In the first inequality, the dependence is on a free parameter $V'$ that must be chosen deterministically, and thus independently of $V_t$, which is a random quantity. The second inequality instead depends directly on the cumulative variance $V_t$, rather than its square root—as ours does—which is necessary for achieving tight regret bounds.
> - [2, Lemma 3]: This provides an inequality that depends on a parameter $\eta$, which (similarly to $V'$ in [1]) must be chosen deterministically and independently of $V_t$. Our bound can be viewed as an extension that allows $\eta$ to depend on the cumulative variance, enabled by the stitching technique.
>
> ### References
>
> [A] Abeille, Marc, Louis Faury, and Clément Calauzènes. "Instance-wise minimax-optimal algorithms for logistic bandits." International Conference on Artificial Intelligence and Statistics. PMLR, 2021.

---

> ### Comment · Reviewer_xX8K · 2025-08-04
>
> Thank you for your responses. They have addressed all my concerns, and I strongly maintain my score of 5.
>
> I have read through all the discussions, especially with Reviewer nkzS. I have some comments of my own on some of the controversies, much of which as comment/question to the authors and some of which as response to Reviewer nkzS.
>
> **Heteroskedasticity**
> - I didn't get the "explicitly told by agent" heteroskedastic noise part, but because we are in a parametric setup (at least in the context of noise distribution), I don't see why this is a problem.
> - $\sqrt{T / \kappa\_*}$ is precisely of the form $\sqrt{\sum\_t \sigma\_t^2}$ of variance-aware linear bandits. Namely, by definition, as the "optimal" variance (at the optimal arm) across the iterations is the same as $\dot\mu(f^\*(x^\*))$, the "analogous" form would be $\sqrt{\sum\_t \dot\mu(f^\*(x^\*))} = \sqrt{T \dot\mu(f^\*(x^\*))} =: \sqrt{T / \kappa\_\*}$.
> - **Question to the authors** Is it straightforward for the results in this paper to be extended to time-varying arm-sets? In that case, one would obtain a regret bound that scales as $\sqrt{\sum\_t \dot\mu(f^\*(x\_t^\*))} = \sqrt{T \sum\_t \frac{1}{T} \dot\mu(f^\*(x\_t^\*))} =: \sqrt{T / \kappa\_\*}$, where $x\_t^\* := \mathrm{argmax}_{x \in X_t} f^*(x)$.
>
> **Regarding the controversy on the effective dimension**, I looked into it myself by checking relevant kernel/neural bandits literature [1,2,3,4,5] as well as a dueling bandit paper [6] and a logistic bandit paper [7]. Here are my opinions:
> - Just in case I was wrong, I , and it seems that the effective dimension is (or should be) non-stochastic, as it takes into account the *entire* context(arm) sets throughout the horizons, and to my knowledge in contextual bandits, the randomness or adversarialness of the arm sets should be independent from the algorithm's randomness. [4, Sec 4.2] first defines this by taking the worst-case over all possible context set sequences; subsequent works [5, Def 3.2] [3, Def 4.3] [1, Def 5.3] doesn't explicitly take the worst-case, but it still constructs (weighted/Hessian-like) design matrix over the entire context set sequence. [1, Def 5.3] even explicitly states "design matrix containing all possible context-arm feature vectors over the $T$ rounds"
> - [2] is a bit confusing, as it states that "Note that placing an assumption on $\tilde{d}$ above is analogous to the assumption on the eigenvalue of the matrix $M_t$ in the work on linear dueling bandits Bengs et al. (2022)," but then [2]'s definition of $\tilde{d}$ (see their Eqn. (4)) seems to be in line with the above; considering the entire context set sequence. I believe this is a typo(..?) on [2]'s side, as Bengs et al. (2022) [6] have two assumptions on the eigenvalues: one critical (Sec. 3.2.1. (A1)) and one not so critical (Corollary 3.3). I believe that [2] was referring to the critical one, which originates from [7], is again independent of the algorithm and is only there to ensure that the context set is sufficiently spanning the whole space.
> - Anyhow, long story short, I agree with nkzS that the authors should be a bit more careful when adding in discussions regarding the effective dimension, as I believe that the effective dimension of prior works do NOT depend on the specific algorithm's randomness. The authors should definitely include more detailed discussions regarding comparison of their information gain to prior notions discussed during the rebuttal.
>
> **Minor stuff**
> - I believe that the authors have indeed cited Srinivas et al. (2010) as [29] (cited in the Appendix footnote 10, but still included in the main text's references)
> - This is something that I missed as well, but as correctly pointed out by nkzS, the authors should take more care in making the exposition more rigorous, especially regarding infinite-dimensional notions such as Fredholm determinants throughout the main text and the proof.
> - (line 245) "The fundamental result in the GLB literature is the bound of [9, Theorem 1]:" => should make it clear that the concentration holds only for Bernoulli.
> - (**should not impact the decision at all**) about a week ago, a preprint (https://arxiv.org/abs/2507.20982) came out that provide some nice overview of the missing references as well as new dimension-free concentration for kernelized logistic regression. For the camera-ready, relevant discussions to this would be nice as well.
>
>
> P.S. I may be wrong on some stuff, especially regarding the quick literature review. Please feel free to let me know if there are anything that I missed or misinterpreted or mistaken in my response. Thanks!
>
>
> [1] https://arxiv.org/abs/2505.02069
>
> [2] https://openreview.net/pdf?id=VELhv9BBfn
>
> [3] https://proceedings.mlr.press/v119/zhou20a/zhou20a.pdf
>
> [4] https://proceedings.mlr.press/v119/yang20h/yang20h.pdf
>
> [5] https://arxiv.org/pdf/2010.00827
>
> [6] https://proceedings.mlr.press/v162/bengs22a/bengs22a.pdf
>
> [7] https://proceedings.mlr.press/v70/li17c/li17c.pdf

---

> > ### Author Response · Authors · 2025-08-04
> >
> > Thank you for supporting our work and for the constructive feedback!
> >
> > ### Heteroskedasticity
> >
> > Thank you for the comment, especially for having highlighted the relationship between $\sqrt{\sum_t \sigma_t^2}$ and $\sqrt{T/\kappa_*}$. We will add a comment in the manuscript to note this point.
> >
> > **Question for the authors:** We believe that, from a technical perspective, the extension to a time-varying arm set does not pose significant challenges. Indeed, the concentration result is not affected by the time-variant decision set, whereas in the regret analysis, it is sufficient to replace $x^\*$ with $x^\*\_t \in \mathcal{X}\_t$, as the Reviewer suggested. Optimism works since the current decision $x\_t$ will be taken from the current decision set $\mathcal{X}\_t$.
> >
> > ### Regarding the controversy on the effective dimension
> >
> > Yes, you are right. The definition of the effective dimension in [4, Sec. 4.2] considers the **sum of all the context/decisions** in the set over the horizon and is therefore not stochastic. Thank you for pointing out our mistake. Instead, our maximal information gain considers the **worst-case sequence of decisions** over the horizon.
> > We will include a detailed comparative discussion in the manuscript, summarizing all the considerations raised during the rebuttal regarding the effective dimension.
> >
> > ### Minor points
> >
> > We will fix the citations and decribe the rigorous treatment for infinite-dimensional notions (e.g., Fredholm determinants). Regarding the recent preprint https://arxiv.org/abs/2507.20982, we came across it and noticed that it addresses a similar problem to ours, concerning the concentration inequality. We will include a comparative discussion in the manuscript.

---

### Official Review · Reviewer_37bp · 2025-07-03

**Clarity:** 3
**Significance:** 3
**Originality:** 3
**Rating:** 5
**Confidence:** 4

**Summary:**

In this paper, the regret minimization problem is studied for Generalized Kernelized Bandits (GKBs) to optimize an unknown function $f^*$ that belongs to a Reproducing Kernel Hilbert Space (RKHS), having access to the noise being generated from an exponential family noise source.  The proposed model is an extension to Kernelized Bandits (KBs) and Generalized Linear Bandits (GLBs). An Upper Confidence Bound (UCB) based algorithm is proposed. As tight regret guarantees can not be obtained using existing self-normalized concentration inequalities, the authors have proposed a novel Bernstein-like dimension-free inequality and claimed that it is a contribution of independent interest. An upper bound on the regret is obtained as a function of learning horizon, maximal information gain, and magnitude of reward nonlinearity. This bound matches the state-of-the-art bounds for KBs and GLBs up to multiplicative constants and logarithmic terms.

**Questions:**

1.	Assumption 3.3 assumes the existence of a bounded noise, which may not hold for practical scenarios, e.g., Gaussian noise. The discussion following Assumption 3.3 needs to be elaborated further for more insight.
2.	Lemma 5.1 reveals that there is a dependence on the maximum slope of the inverse link function and no dependence on the minimum slope. Please discuss why the former is a better thing to happen.
3.	In deriving Theorem 6.2, it is stated that the dependence on $d$ is removed by replacing it with the norm of the feature map, which is bounded under Assumption 3.1? Why can’t we directly assume that $d$ is finite? Would that be a stronger assumption than Assumption 3.1?
4.	The authors are advised to add a small discussion on the heteroscedastic characteristics of the noise for better readability.
5.	For algorithmic and analysis purposes, two different versions of $B_t(\delta;f)$ are used. Moreover, another worst-case version is also introduced. This seems a little confusing. Please clarify what purpose do these different versions serve.
6.	The proposed algorithm can be compared using simulations with suitably adapted versions of the existing algorithms for KBs and GLBs.

**Ethical Concerns:**

["NO or VERY MINOR ethics concerns only"]

**Final Justification:**

The authors have addressed the concerns raised by me. Regarding the over-claim as pointed out by reviewer nkzS, I agree that the contributions of the paper need to stated more clearly without any ambiguity. As stated by reviewer xX8K, I also believe the authors have addressed the main concerns during the rebuttal. I am happy to raise my score to 5.

**Limitations:**

Yes

**Quality:**

3

**Strengths And Weaknesses:**

The paper is well-written and technically solid. However, more intuitions need to be provided to understand the impact of the result better. Moreover, at some places, it is difficult to follow due to deferring some of the discussions towards the end and insufficient details regarding a few important concepts. No simulation results are presented.

---

> ### Author Rebuttal · Authors · 2025-07-31
>
> We thank the Reviewer for the feedback and for appreciating the technical solidity of the paper. In the final version, we will make use of the additional page to introduce the discussion earlier and provide more details on the important concepts. Below, we provide answers to the Reviewer's questions.
>
> 1. We believe that this assumption is quite mild. As we discuss immediately below the assumption, for $\nu^2$-sub-Gaussian noise $\epsilon_t$ (which includes Gaussian noise), we can always set $R = \nu\sqrt{2\log(2T/\delta)}$, which guarantees that $|\epsilon_t| \le R$ *uniformly* for all $t \in [T]$ with probability at least $1 - \delta$. This follows from a union bound argument:
> $$
> \begin{aligned}
> \Pr\left(\forall t \in [T]: |\epsilon_t| \le \nu\sqrt{2\log(2T/\delta)}\right)
> & \ge 1- \Pr\left(\exists t \in [T]: |\epsilon_t| > \nu\sqrt{2\log(2T/\delta)}\right) \\\\
> & \ge 1 - \sum_{t \in [T]} \Pr\left(|\epsilon_t| > \nu\sqrt{2\log(2T/\delta)}\right) \\\\
> & \ge 1- \sum_{t \in [T]} \frac{\delta}{T} = 1-\delta,
> \end{aligned}
> $$
> where the second inequality follows from the union bound, and the last one from Hoeffding's inequality for sub-Gaussian random variables.
> We will include this elaboration in the final version.
>
> 2. This is desirable because many link functions used in practice, such as the sigmoidal one, have a bounded maximum slope (e.g., $R_{\dot{\mu}} = 1$ for the sigmoid and the identity function), while their minimum slope can approach zero (i.e., $\kappa_{\mathcal{X}} \approx 0$ for the sigmoid function). Therefore, a dependence on $1/\kappa_{\mathcal{X}}$, as found in early works on generalized linear bandits [e.g., A], is undesirable.
>
> 3. In a kernel space, the dimensionality $d$ of the feature map $\phi$ induced by the kernel $k$ can be infinite. Thus, assuming $d$ to be finite would reduce the setting to the already studied generalized linear bandit case. Moreover, even under the assumption of finite $d$, it would not be possible to remove Assumption 3.1, which—in the generalized linear bandit case—reduces to requiring a bounded norm of the parameter $\theta$ [see B, Assumptions 1 and 2].
>
> 4. The heteroscedasticity of the noise is directly related to the noise model, which is based on a **canonical exponential family**. Under this model, the variance of the noise is proportional to the slope of the link function, i.e., $\text{Var}[\epsilon_t] \propto \dot{\mu}(f(x))$. We will emphasize this point in the final version.
>
> 5. We introduce two versions, $B_t(\delta, f)$ and $\beta_t(\delta, f)$, which differ in the use of the information gain: $\widetilde{\Gamma}_t(f)$ versus the maximal information gain $\widetilde{\gamma}_t(f)$. Consequently, $B_t(\delta, f) \le \beta_t(\delta, f)$, where the former is a random variable and the latter is a deterministic quantity.
> Moreover, we consider their worst-case versions over the choice of $f$: namely, $B_t(\delta, \mathcal{H})$ and $\beta_t(\delta, \mathcal{H})$, respectively. These quantities satisfy the following relations:
> $$
> B_t(\delta,f) \le \begin{cases} \beta_t(\delta,f)  \\\\ B_t(\delta,\mathcal{H}) \end{cases} \le \beta_t(\delta,\mathcal{H}).
> $$
> In principle, we could use the largest one, $\beta_t(\delta, \mathcal{H})$, from the beginning. However, using the former ones allows us to obtain tighter bounds, which improve the theoretical guarantees and may also lead to practical improvements. We will include a clarification on that in the paper.
>
> 6. Although we believe that the core contributions of the paper are mainly theoretical and algorithmic, we agree that some numerical simulations comparing the two versions of our algorithm, as well as comparisons with existing ones for KBs and GLBs, would be of interest. We are running these simulations, and we will include them in the final version of the paper.
>
> ### References
>
> [A] Jun, Kwang-Sung, et al. "Scalable generalized linear bandits: Online computation and hashing." Advances in Neural Information Processing Systems 30 (2017).
>
> [B] Abeille, Marc, Louis Faury, and Clément Calauzènes. "Instance-wise minimax-optimal algorithms for logistic bandits." International Conference on Artificial Intelligence and Statistics. PMLR, 2021.

---

> > ### Comment · Reviewer_37bp · 2025-08-05
> >
> > I have read the responses provided by the authors and comments by the reviewers. The authors have addressed the concerns raised by me. Regarding the over-claim as pointed out by reviewer nkzS, I agree that the contributions of the paper need to stated more clearly without any ambiguity. As stated by reviewer xX8K, I also believe the authors have addressed the main concerns during the rebuttal. Numerical simulations conducted by the authors should be included in the final version.

---

> > > ### Author Response · Authors · 2025-08-05
> > > **Re: Official Comment by Reviewer 37bp**
> > >
> > > Thank you for the feedback. We are happy to have addressed your concerns. In the final version, we commit to fixing the statements about the contributions to avoid any ambiguity, and we will include the numerical simulation.

---

### Official Review · Reviewer_nkzS · 2025-07-03

**Clarity:** 3
**Significance:** 2
**Originality:** 2
**Rating:** 2
**Confidence:** 4

**Summary:**

The authors proposed a new nonparametric form subsuming both generalized linear models and function approximation in reproducing kernel Hilbert spaces. Under this framework of general function approximation, they adapted weighted ridge regression to this bandit setting so as to achieve a $\sqrt{T}$ high-probability regret bound using optimism in the face of uncertainty. The authors also came up with an efficient approximation of their algorithm.

**Questions:**

$\gamma_T$ in this paper is essentially the same notion as the "effective dimension" in the neural bandits literature like [1] and corresponds to the dimension $d$ in the linear bandits literature, as stated in [2], right?

### References

[1] Zhou, D., Li, L., & Gu, Q. (2020, November). Neural contextual bandits with ucb-based exploration. In International Conference on Machine Learning (pp. 11492-11502). PMLR.

[2] Srinivas, N., Krause, A., Kakade, S. M., & Seeger, M. (2009). Gaussian process optimization in the bandit setting: No regret and experimental design. arXiv preprint arXiv:0912.3995.

> By the way, the missing citation of [2] is really not as expected, because this GP-UCB paper proposes the $\gamma_T$ for the first time in the bandit community.

**Ethical Concerns:**

["NO or VERY MINOR ethics concerns only"]

**Final Justification:**

I have an in-depth discussion with the authors regarding their contributions. I believe that I have fully understood what the authors **try to** claim. However, I found that **every concern (about their contributions) I raised does hold water** and the best the authors have done is mainly reiterating what I concern about by stating "what we intended by ... is ..." or "this is what we intended by ...". Therefore, my conclusion is that the contribution of this submission is nearly not novel and **Table 1 in this submission contains significant overclaims**.
- ***I am willing to discuss about this submission with the area chair*** if necessary.

**Limitations:**

Yes.

**Paper Formatting Concerns:**

N/A.

**Quality:**

2

**Strengths And Weaknesses:**

### Strengths

- Most mathematical derivations are clear and easy to follow.
- Theorem 6.2 and its proof align with the intuition of the community of variance-aware online learning.
- The effort in deriving the efficient implementation in Appendix A sets a good role model for the theory community.

### Weaknesses on Credit Assignment and Over-claiming

- At the message and technical level, the idea of "kernel + link function" already appears in the bandit literature, e.g., [1]. In particular, [1] achieved nearly the same regret bound as that in this paper using similar (and even simpler) technical components, in which their "effective dimension" is essentially the same as the author's "maximal information gain". [1] appeared on arXiv about a week before the submission deadline and the authors did not discuss it in the paper.
- Table 1 is made to show that this paper is the only one that can tackle heteroskedastic noise besides the work [2]. However, what the authors actually considered is the "variance-revealing" case, in which the potentially heteroskedastic variance of the picked arm is given to the learner. When $\mu$ and $f^*$ are unknown, such an implicit assumption is not entirely reasonable. In contrast, certain papers on linear and generalized linear bandits back to two years again, e.g., [3] and [4], are already able to handle heteroskedastic noise without such type of variance-revealing assumption.
- Both quantities appear in Assumption 3.1 and Assumption 3.2 should not be treated as "constants". In particular, the norm bouund $B$ and the kernel bound $K$ are important problem-dependent quantities. Therefore, "state-of-the-art" statement in the Abstract of this paper is an **over-claiming** because the dependency on $B$ and $K$ is not optimal when specialized to generalized linear bandits, given the true SoTA [5], which the authors also mentioned in the conclusion.
- The authors called the standard peeling technique "stitching" and claimed its application to deriving "empirical" Freedman to be "of independent interest". However, this type of peeling in improving Freedman has been widely used in the variance-aware online learning community. To be concrete, Theorem 9 of [6] and Appendix A of [7] are two examples of such peeling techniques.

### Weaknesses on Rigorousness and Correctness

- When deriving equation (68), which is a key part in proving Theorem 6.2, the authors invoked *their version of elliptical potential lemma*, Lemma C.6. In the last step of deriving Lemma C.6, the authors claimed that is follows from the proof in [8]. However, if we truely follow the derivation of Lemma 12 of [8], two issues arise:
  1. Even in the finite-dimensional case, the term $\lambda^{-1}$ on the right-hand side of equation (196) does not really align with $(\det(\lambda I))^{-1}$ if the dimension is not one.
  2. To make matters worse, in the infinite-dimensional case, there is in general no well-defined determinant for an operator. Technically speaking, the only known and widely used notion of determinant for infinite-dimensional real Hilbert space is the Fredholm determinant, which is only applicable for operators of the for $I + A$ where $\sqrt{A^\top A}$ has a finite trace. See, e.g., https://en.wikipedia.org/wiki/Fredholm_determinant for details. Therefore, even though equation (196) might intuitively make sense for infinite-dimensional Hilbert spaces, it is not mathematically clear what the authors actually did to achieve the $\leq$ in equation (196). I suggest the authors to first try to write down **the concrete definition** of determinant of some necessarily involved operators in deriving this inequality (196), instead of citing [8] in a sloppy way.

#### References

[1] Bae, S., & Lee, D. (2025). Neural Logistic Bandits. arXiv preprint arXiv:2505.02069. (https://arxiv.org/pdf/2505.02069)

[2] Faury, L., Abeille, M., Calauzènes, C., & Fercoq, O. (2020, November). Improved optimistic algorithms for logistic bandits. In International Conference on Machine Learning (pp. 3052-3060). PMLR.

[3] Zhao, H., He, J., Zhou, D., Zhang, T., & Gu, Q. (2023, July). Variance-dependent regret bounds for linear bandits and reinforcement learning: Adaptivity and computational efficiency. In The Thirty Sixth Annual Conference on Learning Theory (pp. 4977-5020). PMLR.

[4] Di, Q., Jin, T., Wu, Y., Zhao, H., Farnoud, F., & Gu, Q. (2023). Variance-aware regret bounds for stochastic contextual dueling bandits. arXiv preprint arXiv:2310.00968.

[5] Lee, J., Yun, S. Y., & Jun, K. S. (2024). A unified confidence sequence for generalized linear models, with applications to bandits. arXiv preprint arXiv:2407.13977.

[6] Zimmert, J., & Lattimore, T. (2022, June). Return of the bias: Almost minimax optimal high probability bounds for adversarial linear bandits. In Conference on Learning Theory (pp. 3285-3312). PMLR.

[7] [Li, G., Cai, C., Chen, Y., Wei, Y., & Chi, Y. (2021). Is Q-Learning Minimax Optimal? A Tight Sample Complexity Analysis. arXiv preprint arXiv:2102.06548.](https://arxiv.org/pdf/2102.06548)

[8] Abeille, M., Faury, L., & Calauzènes, C. (2021, March). Instance-wise minimax-optimal algorithms for logistic bandits. In International Conference on Artificial Intelligence and Statistics (pp. 3691-3699). PMLR.

---

> ### Author Rebuttal · Authors · 2025-07-31
>
> We thank the Reviewer for the feedback and for appreciating the clarity of the mathematical derivations and the effort in deriving an efficient implementation. Below, we address the Reviewer's concerns.
>
> ### Weaknesses on Credit Assignment and Over-claiming
>
> - We thank the Reviewer for reference [1]. First of all, we note that, according to the **NeurIPS Call for Papers**, [1], being made available online after March 1st, **must be considered contemporaneous work**. Therefore, the presence of [1] cannot be used as an argument to diminish the novelty of our paper.  Nevertheless, we provide below a comparative discussion with our work, which we will include in the final version of the paper:
> 1. Section 3 of [1] provides a concentration bound (Theorem 3.1) similar to our Theorem 6.2. However, their result (i) assumes a *finite-dimensional* decision set; and (ii) exhibits a worse dependence on $t$, being of order $\log(t)$, whereas ours is of order $\log\log(t)$.
> Looking at Appendix A of [1], it can be observed that their proof technique is very similar to ours, as both are inspired by Zhou et al. (2022). Thus, [1] is *not using simpler techniques*, except for avoiding the stitching/peeling argument, which we specifically use to reduce the $\log(t)$ dependence to $\log\log(t)$.
>
>
> 2. The *effective dimension* $\widetilde{d}$ in Definition 5.3 of [1] is **not** the same as our *maximal information gain* $\gamma_T$. Indeed, Definition 5.3 of [1] does not consider the exact variance term $\dot{\mu}(f^*(x_t))$, which is instead upper bounded by its maximum value $R$ (denoted as $R_{\dot{\mu}}$ in our notation).
> In contrast, in our definition in Section 5, we retain the variance terms $\dot{\mu}(f(x_t)) \le R$ within the definition of *information gain*, which leads to the bound $\widetilde{\Gamma}_t(f) \le \widetilde{d}$. Even when considering the worst-case sequence of decisions, leading to the maximal information gain $\gamma_T$, we still retain the variance factors. Finally, $\widetilde{d}$ is a **random quantity that depends on the specific realization of the algorithm's execution** (i.e., the sequence of decisions). For this reason, it is **inappropriate to present a regret bound, such as in Theorem 5.5 of [1], that depends on it**. Instead, a maximal version based on the worst-case sequence of decisions should be considered (like $\gamma_T$), as is standard in approaches like [2].
>
>
> 3. The regret bound of [1, Theorem 7.2] holds under Condition 5.4, which in turn involves several assumptions (5.1 and 5.2) that are not required by our algorithm. Before comparing the regret bounds, a few considerations:
>
> (i) [1, Theorem 7.2] does not show a dependence on the norm of the decisions $\\|x\\|_2$, since under their Assumption 5.2 it is assumed to be 1. This corresponds, in our notation, to setting $K=1$;
>
> (ii) [1, Theorem 7.2] depends on $S$, which is defined in Definition 6.2 and is therefore related to the bound on the norm of the parameters $\theta$. Thus, apart from constants (like $m$, the number of hidden neurons in the network), $S$ corresponds to our parameter $B$.
>
> Given these considerations, we can effectively compare our regret bound with that of [1] (setting $K=1$ and assuming $\widetilde{d} \approx \widetilde{\gamma}_T$ for a fair comparison):
>
> \begin{align}
> \\text{Our bound:} \\qquad \\widetilde{O}\\left(  B \\widetilde{\\gamma}\_T \\sqrt{\\frac{T}{\\kappa\_*}} + B^2 \sqrt{\frac{\widetilde{\gamma}\_T T }{{\kappa\_{\*}}}} \\right).
> \end{align}
>
> \begin{align}
> \text{Bound of [1, Theorem 7.2]:} \qquad \widetilde{O}\left(B^2 \widetilde{\gamma}\_T \sqrt{\frac{T}{\kappa\_{\*}}} + B^{2.5} \sqrt{\frac{\widetilde{\gamma}\_T T}{\kappa\_{\*}}} \right).
> \end{align}
>
> Thus, our bound exhibits a better dependence on $B$.
>
>
> - Regarding the heteroschedasticity, the goal of our Table 1 is to compare the approaches able to tackle **generalized linear bandits** or **kernelized bandits**. We thank the Reviwer for the references, however, we think they do not consider  these settings. Indeed, [3] focuses on dueling bandits that differ from our setting from the interaction protocol (although a link function is present as well), whereas [4] focuses on linear bandits. Nevertheless, we will cite them in the related works section to mention further works that focus on the heteroschedastic case.
>
> - As the Reviewer noted, we acknowledged the non-tight dependence on $K$ and $B$ in the conclusions. In the abstract, we intended to specify the tightness of the dependence only with respect to $\gamma_T$, $T$, and $\kappa_*$.
> To avoid any risk of overclaiming, we will rephrase the sentence in the abstract as:
> *"Our results match, up to multiplicative constants and logarithmic terms, the state-of-the-art bounds for both KBs and GLBs **with respect to $\gamma_T$, $T$, and $\kappa_*$, while showing suboptimal dependence on $B$ and $K$**."*
>
>
>
> - First of all, we clarify that the term **stitching** is widely used in the community dealing with confidence sequences [e.g., A, B].
> More importantly, **we never claimed that our empirical Freedman inequality in Theorem 6.1 is of independent interest**. Indeed, we are aware that similar results already exist, and **this is evidenced by the sentence between lines 282 and 283, where we cite a result, [12, Theorem 12], analogous to those provided by the Reviewer (Theorem 9 of [6] and Appendix A of [7])**.
> We will include the references suggested by the Reviewer in the final version. However, we highlight that they compare analogously to [12, Theorem 12] in relation to our inequality. Specifically, as stated between lines 283–284, *"our result allows tuning the parameters $\eta$ and $v_0$ to tighten the bound, ultimately leading to an improvement of the constants."*
> **We emphasize once again that this is the only improvement over the state of the art that we claim for Theorem 6.1, namely, generality in $\eta$ and $v_0$, and better constants** ($\sqrt{2}$ in our case, compared to $3$ in Theorem 9 of [6], and $\sqrt{8}$ in Appendix A of [7]). We will further clarify this in the final version.
> The phrase **independent interest** appears in our manuscript at lines 10 and 346, and in both cases, it unequivocally refers to the **self-normalized bound of Theorem 6.2**, which is novel.
>
>
> ### Weaknesses on Rigorousness and Correctness
>
>
> Thank you for pointing out the issue. As we mentioned in Footnote 12, we focused on the setting of paper [32], as our derivations are valid under Assumptions 1 and 2 of [32], which are the same as those considered in our paper.
> As the Reviewer correctly points out, in [32], the determinant used is the Fredholm determinant. We will clarify this in our manuscript.
> Regarding the specific derivation in line (196), we note that the same argument (in the infinite-dimensional case) is employed in the "Proof Sketch for Theorem 2," which relies on Lemma 5 of [32], a result that holds in infinite-dimensional Hilbert spaces.
> For this reason, we will replace the citation to [8] with a reference to [32, Lemma 5] and mention that we are working with Fredholm determinants.
>
>
> ### References
>
> [1] Bae, S., & Lee, D. (2025). Neural Logistic Bandits. arXiv preprint arXiv:2505.02069.
>
> [2] Faury, L., Abeille, M., Calauzènes, C., & Fercoq, O. (2020, November). Improved optimistic algorithms for logistic bandits. In International Conference on Machine Learning (pp. 3052-3060). PMLR.
>
> [3] Zhao, H., He, J., Zhou, D., Zhang, T., & Gu, Q. (2023, July). Variance-dependent regret bounds for linear bandits and reinforcement learning: Adaptivity and computational efficiency. In The Thirty Sixth Annual Conference on Learning Theory (pp. 4977-5020). PMLR.
>
> [4] Di, Q., Jin, T., Wu, Y., Zhao, H., Farnoud, F., & Gu, Q. (2023). Variance-aware regret bounds for stochastic contextual dueling bandits. arXiv preprint arXiv:2310.00968.
>
> [5] Lee, J., Yun, S. Y., & Jun, K. S. (2024). A unified confidence sequence for generalized linear models, with applications to bandits. arXiv preprint arXiv:2407.13977.
>
> [6] Zimmert, J., & Lattimore, T. (2022, June). Return of the bias: Almost minimax optimal high probability bounds for adversarial linear bandits. In Conference on Learning Theory (pp. 3285-3312). PMLR.
>
> [7] Li, G., Cai, C., Chen, Y., Wei, Y., & Chi, Y. (2021). Is Q-Learning Minimax Optimal? A Tight Sample Complexity Analysis. arXiv preprint arXiv:2102.06548.
>
> [8] Abeille, M., Faury, L., & Calauzènes, C. (2021, March). Instance-wise minimax-optimal algorithms for logistic bandits. In International Conference on Artificial Intelligence and Statistics (pp. 3691-3699). PMLR.
>
> [A] Kuchibhotla, Arun K., and Qinqing Zheng. "Near-Optimal Confidence Sequences for Bounded Random Variables." International Conference on Machine Learning. PMLR, 2021.
>
> [B] Chugg, Ben, Hongjian Wang, and Aaditya Ramdas. "Time-Uniform Confidence Spheres for Means of Random Vectors." Transactions on Machine Learning Research.
>
> [32] Justin A. Whitehouse, Aaditya Ramdas, and Zhiwei S. Wu. On the sublinear regret of GP-UCB. In Advances in Neural Information Processing Systems (NeurIPS), volume 36, pages 35266–35276, 2023.

---

> > ### Comment · Reviewer_nkzS · 2025-07-31
> >
> > I have carefully reviewed the rebuttal from the authors, especially regarding the elliptical potential part. Base on the rebuttal, could I conclude that the self-normalized concentration inequality in this submission is dimension-free only if the underlying Hilbert space is of countably **infinite** dimension?
> > - I raise this question because, in Table 1 of this submission, some self-normalized bounds in previous works for **finite**-dimensional linear bandits are classified as "not dimension-free". But if the main concentration inequality in this submission is also not dimension-free in the **finite**-dimensional setting, the comparison in Table 1 might be unfair.

---

> > > ### Author Response · Authors · 2025-08-01
> > > **Re: Official Comment by Reviewer nkzS**
> > >
> > > Thank you for carefully reviewing our rebuttal. Yes, indeed, our self-normalized concentration inequality holds for RKHSs with a **countably infinite** basis. To clarify, by **dimension-free**, we mean that the bound does not depend on the dimensionality of the space, that is, it does not depend on the cardinality $d$ of a minimal basis $\\{\phi\_i\\}\_i$.
> > > We emphasize that this holds in both the **countably infinite** and **finite-dimensional** cases. In fact, our self-normalized concentration inequality depends only on the norm of the feature vector only $
> > > K = \sup_{x \in \mathcal{X}} \\|\phi(x)\\|\_2 = \sup_{x \in \mathcal{X}} k(x,x),
> > > $
> > > and not on $d$.
> > > In contrast, for example, the bounds in Faury et al. (2020) [9, Theorem 1] and Zhou et al. (2021) [36, Theorem 4.1] include an explicit dependence on $d$. We will clarify this point in the final version.

---

> ### Comment · Reviewer_nkzS · 2025-08-01
>
> But in the finite-dimensional setting, no matter you are using [32, Lemma 5] or [8, Lemma 12], how could the $\log\det$ stuff in the proof be dimension-free?
> - If you use [8, Lemma 12] directly in the finite-dimensional setting, there should be a linear dependency on $d$, right?
> - If you use [32, Lemma 5], how could it be applicable for the finite dimensional setting?
>
> To be more concrete, in the finte-dimensional setting, $\det(\lambda I)$ is $\lambda^d$ and $\log (\det (\lambda I))^{-1}$ will be linear in $d$.
>
> ---
>
> References
>
> [8] Abeille, M., Faury, L., & Calauzènes, C. (2021, March). Instance-wise minimax-optimal algorithms for logistic bandits. In International Conference on Artificial Intelligence and Statistics (pp. 3691-3699). PMLR.
>
> [32] Justin A. Whitehouse, Aaditya Ramdas, and Zhiwei S. Wu. On the sublinear regret of GP-UCB. In Advances in Neural Information Processing Systems (NeurIPS), volume 36, pages 35266–35276, 2023.

---

> ### Author Response · Authors · 2025-08-01
> **Re: Official Comment by Reviewer nkzS**
>
> Thank you for your answer, your concern is now clearer to us. We take this opportunity to further clarify:
>
> - When claiming the **dimension-free** property, we were referring to the **explicit dependence on $d$** in the concentration inequality. Our Theorem 6.2 does not exhibit such a dependence.
>
> - The Reviewer is referring, instead, to an **implicit dependence on $d$**, which appears in the **bound** of the log-determinant term $\log \det (\lambda^{-1}\widetilde{V}_t(\lambda))$ in Theorem 6.2. Specifically,
>   $$
>   \log \det (\lambda^{-1}\widetilde{V}_t(\lambda)) = \log \det (I + \lambda^{-1} \widetilde{V}_t) \le d \log \left( 1 + \frac{K^2 R\_{\dot{\mu}} }{\mathfrak{g}(\tau) \lambda d} (t-1) \right),
>   $$
>   which can be derived analogously to Lemma C.7. This bound applies to **finite-dimensional** feature spaces; but other bounds for the term $\log \det (\lambda^{-1}\widetilde{V}_t(\lambda))$ for example, using the *information gain*, are also possible and meaningful even in the finite-dimensional case.
>
> - Nevertheless, for *generalized linear bandits*, it has been shown that dependence on the dimensionality $d$ is **unavoidable in the worst-case scenario** (see, for instance, Theorem 2 of [2]). This is the reason why the bound to $\log \det (\lambda^{-1}\widetilde{V}_t(\lambda))$ contains a dependence on $d$ and, in this sense, **cannot be dimension-free**.
>
> - Finally, we remark that, even the seminal inequality of Abbasi-Yadkori et al. (2011) [1], **which we also classified as "dimension-free" in Table 1**, contains a log-determinant term, whose bound depends on $d$ (see Theorem 2 of [1] statements 1 and 2).
>
> We hope this clarifies the issue. We will make sure to include this discussion on this point.
>
> ### References
>
> [1] Yasin Abbasi-Yadkori, Dávid Pál, and Csaba Szepesvári. Improved algorithms for linear stochastic bandits. In Advances in Neural Information Processing Systems (NIPS), pages 2312–2320, 2011
>
> [2] Marc Abeille, Louis Faury, and Clément Calauzènes. Instance-wise minimax-optimal algorithms for logistic bandits. In International Conference on Artificial Intelligence and Statistics (AISTATS), pages 3691–3699. PMLR, 2021.

---

> ### Comment · Reviewer_nkzS · 2025-08-01
>
> Let us say just for linear logistic bandits with feature dimension $d$. No matter how you redefine the $\log\det$ stuff into new quantities like "maximal information gain", both the self-normalized concentration bounds in this submission and [F] scale with $\mathrm{poly}(d)$, right? But [F] is classified as "not dimension-free" in Table 1 of this submission. Is that my misunderstanding???
>
> ---
>
> References
>
> [F] Faury, L., Abeille, M., Calauzènes, C., & Fercoq, O. (2020, November). Improved optimistic algorithms for logistic bandits. In International Conference on Machine Learning (pp. 3052-3060). PMLR.

---

> ### Author Response · Authors · 2025-08-02
> **Re: Official Comment by Reviewer nkzS**
>
> First of all, let us remark that the reason why we classified [F] as "non-dimension-free" is because of the **explicit dependence** on $d$ in the last addendum of Theorem 1: $ \frac{2}{\sqrt{\lambda}} \boldsymbol{\color{red}d} \log(2)$ and **not** because the log-determinant is bounded by to $\text{poly}(d)$. That said, [F] and our inequality, **when restricted to logistic bandits**, contain precisely the same log-determinant term: $\log \det (\lambda^{-1} \widetilde{V}\_t(\lambda)) =  \log \det (I + \lambda^{-1} \widetilde{V}_t)$ with $\\widetilde{V}\_t = \\sum\_{s=1}^{t-1} \\dot{\\mu}(\\theta^\\top x\_s) x\_s x\_s^\\top$.
>
> Now, the question becomes whether $\log \det (\lambda^{-1} \widetilde{V}\_t(\lambda))$  is always $\text{poly}(d)$ , or only bounded to $\text{poly}(d)$ in certain cases. Consider the following example, where we assume $\\dot{\\mu}(\\theta^T x\_s) = 1$ for simplicity (i.e., linear bandits).  Suppose $x_s = (1,0,\dots,0)^\\top$ for every $s \\in \\{1,\dots,t-1\\}$, we have that:
> $$\\widetilde{V}\_t = \\begin{pmatrix} t-1 & 0 & \\dots & 0 \\\\ 0 & 0 & \dots & 0 \\\\ \\vdots & \\vdots & \\ddots & \\vdots \\\\ 0 & 0 & \dots & 0  \\end{pmatrix}, \qquad \\lambda^{-1} \\widetilde{V}\_t(\lambda) = I + \\lambda^{-1} \\widetilde{V}\_t =   \\begin{pmatrix} 1 + \\lambda^{-1}(t-1) & 0 & \\dots & 0 \\\\ 0 & 1 & \dots & 0 \\\\ \\vdots & \\vdots & \\ddots & \\vdots \\\\ 0 & 0 & \dots & 1  \\end{pmatrix}.$$
> As a consequence, $\log \det (\lambda^{-1} \widetilde{V}\_t(\lambda))  = \log(1 + \\lambda^{-1}(t-1) )$, which **shows no dependence on $d$**. Clearly, this is a rather unrealistic case when applying our inequality or that of [F] in a regret minimization algorithm. Still, restricted to the discussion of the properties of the concentration inequality, this provides an example in which our inequality shows **no dependence on $d$ at all**, whereas that of [F] does, due to the presence of the term $ \frac{2}{\sqrt{\lambda}} \boldsymbol{\color{red}d} \log(2)$.
>
> Nevertheless, if the Reviewer believes that our definition of **dimension-free** as *lack of explicit dependence on $d$* may be misleading, we are open to changing it (including also *implicit dependencies on $d$*) in the manuscript to avoid any confusion.

---

> ### Comment · Reviewer_nkzS · 2025-08-03
>
> I have read the authors elaboration on what they intend to claim by "dimension-free" and believe I have understand what they want to claim about it. Regarding other technical parts of the rebuttal. I have the following viewpoints as a follow-up.
>
> 1. Based on my understanding, in the rebuttal and subsequent responses, the authors try to battle with themselves as follows.
>     - The authors consider it to be inappropriate the place the executation-dependent quantity $\\tilde{d}$, which is termed as the "effective dimension" in standard neural bandits literature [Z, V, 1], in the upper bounds and claim that their worst-case quantity $\gamma_T$ (the maximal information gain) to be more appropriate than that in [1] when comparing these two concurrent works.
>     - On the other hand, when the authors discuss about their "worst-case $\mathrm{poly}(d)$" quantity $\\log\\det(\\lambda^{-1}\tilde{V}\_t(\\lambda))$, they try to justify that this **executation-dependent** quantity can be dimension-free when the executation has nice $\\{x_s\\}\_{s=1}^{t-1}$.
>     - Overall, I suggest the authors to have a consistent tone when discussing their work and others. By the way, a minor point is that, if it is indeed the case that $\tilde{d} \lesssim \gamma_T$, it is not very appropriate to claim the introduction of $\gamma_T$ to be a key difference of that $\tilde{d}$ in [1].
> 2. In the authors' rebuttal bullet "Regarding the heteroschedasticity ...", there are several points to remind.
>     - The authors claim that "[3] focuses on dueling bandits", which is not true. [3] focuses on linear bandits.
>     - Also, as a minor point, at least two papers on linear bandits appear in Table 1 of this submission, but in the rebuttal, the  authors state that "the goal of our Table 1 is to compare the approaches able to tackle generalized linear bandits or kernelized bandits".
>     - **Most importantly**, the authors claim that they method can tackle heteroscedastic noise, but **only when** the heteroscedastic variance is **explicitly told to the agent**. This is considered a significant drawback because such a "explicit variance revealing" is already not needed for the linear and generalized linear settings, at least several years ago, see, e.g., [3,4].
>        - I definitely know that "dueling ... are not exactly genearlized linear ..." and "kernel ... is not exactly linear ...", but technically speaking, since "explicit varaince revealing" can be avoided in linear bandits or bandits with link functions; it should be, mathematically, not needed for at least the finite-dimensional case in this submission.
>        - Since "heteroscedasticity" is really an application-inspired concept, it is not very clear why could the agent see the heteroscedastic variance by explicitly know $g(\tau)\dot{\mu}(f(\mathbf{x}))$, where we know that the noise variance is exactly $g(\tau)\dot{\mu}(f(\mathbf{x}))$.
>
> ---
>
> References
>
> [Z] Zhou, D., Li, L., & Gu, Q. (2020, November). Neural contextual bandits with ucb-based exploration. In International conference on machine learning (pp. 11492-11502). PMLR.
>
> [V] Verma, A., Dai, Z., Lin, X., Jaillet, P., & Low, B. K. H. (2024, January). Neural dueling bandits. In ICML 2024 Workshop: Foundations of Reinforcement Learning and Control--Connections and Perspectives.
>
> [1] Bae, S., & Lee, D. (2025). Neural Logistic Bandits. arXiv preprint arXiv:2505.02069.
>
> [3] Zhao, H., He, J., Zhou, D., Zhang, T., & Gu, Q. (2023, July). Variance-dependent regret bounds for linear bandits and reinforcement learning: Adaptivity and computational efficiency. In The Thirty Sixth Annual Conference on Learning Theory (pp. 4977-5020). PMLR.
>
> [4] Di, Q., Jin, T., Wu, Y., Zhao, H., Farnoud, F., & Gu, Q. (2023). Variance-aware regret bounds for stochastic contextual dueling bandits. arXiv preprint arXiv:2310.00968.

---

> > ### Comment · Reviewer_nkzS · 2025-08-03
> >
> > Another follow-up question is, if the authors claim that the algorithm can deal with heterscedastic variance, why the final regret bound seems like a worst-case one? In other words, the regret bounds is not something proportional to $\sqrt{\sum_{t} \sigma_t^2}$.

---

> > > ### Author Response · Authors · 2025-08-03
> > > **Re: Official Comment by Reviewer nkzS**
> > >
> > > We are happy that we have clarified what we intended by "dimension-free." Regarding the follow-up questions:
> > >
> > > 1. We believe that we kept a *consistent tone* in discussing $\widetilde{d}$, $\gamma_T$, and $\log\det(\lambda^{-1}\tilde{V}_t(\lambda))$.
> > >
> > >     * Yes, it is more appropriate in our view to have a decision-independent quantity like $\gamma_T$ in the regret bound since it is not random and allows us to quantify the (high-probability) regret of the algorithm before executing it, i.e., before seeing the sequence of decisions. **This is in line with the linear bandit, generalized linear bandit, and kernelized bandit literature [i, Theorem 4], [ii, Theorem 1], [iii, Theorem 2], [iv, Theorems 2 and 3]. We don't see the point of deviating from such a practice.** Since the goal of this work is to combine generalized linear bandits and kernelized bandits, we followed the established practices in these areas to enable a fair comparison.
> > >
> > >     * We indeed provided an **explicit example** in which $\log\det(\lambda^{-1}\tilde{V}_t(\lambda))$ is not dependent on $d$.
> > >
> > >     * **The two points above are consistent.** The first one is about the regret bound, and the second one is about the algorithm. Indeed, using $\log\det(\lambda^{-1}\tilde{V}_t(\lambda)) \le \widetilde{d}$ in the confidence radius limits the exploration of the algorithm without compromising its theoretical guarantees. **So, our desideratum is: use the smallest (random) confidence radius in the algorithm, $\log\det(\lambda^{-1}\tilde{V}_t(\lambda)) \le \widetilde{d}$, and then present the regret bound as a deterministic expression, i.e., $\gamma_T$, for the reasons clarified above.**
> > >
> > > 2. About heteroscedasticity:
> > >
> > >     * Of course, it is a typo. We misplaced reference [3] with reference [4], as this is apparent in the sentence "Indeed, [3] focuses on dueling bandits that differ from our setting from the interaction protocol (although a link function is present as well), whereas [4] focuses on linear bandits."
> > >
> > >     * Yes, the reviewer is right. The first two papers in Table 1 refer to linear bandits since they are the ones on top of which both generalized linear bandits and kernelized bandits build.
> > >
> > >     * We take the opportunity to clarify the origin of the stochasticity in our work. We remind that we are considering a **canonical exponential family** noise model (Equation 2), which, for some choices of its parameters, implicitly generates noise random variables with different variances depending on the decision $x$. Because of the properties of the canonical exponential family, the variance of the noise is directly related to its parameters $\mathfrak{g}(\tau) \dot{\mu}(f(x))$. Given this, some observations are in order:
> > >
> > >         * The function $\mathfrak{g}(\tau) \dot{\mu}(\cdot)$ is known to the agent, as in all literature about generalized linear bandits [e.g., ii, iv], and **as a consequence of this, the variance of the noise is revealed to the agent**. **This is what we intend by heteroscedasticity. If the Reviewer thinks that this term is inappropriate and/or misleading, we are open to any suggestion to change it. Still, it remains the common practice in the works on generalized linear bandits.**
> > >
> > >         * The Reviewer suggests that "since "explicit varaince revealing" can be avoided in linear bandits or bandits with link functions; it should be, mathematically, not needed for at least the finite-dimensional case in this submission." First of all, paper [3] considers Bernoulli feedback, which is a specific case of the canonical exponential family. Even assuming, following the Reviewer's claim, that this problem is solved in bandits with link function (we think that it might be the case for logistic ones or, at least, for ones with Bernoulli observations, but not in general for the canonical exponential family), it remains an open problem in kernelized bandits, as far as we know. **Thus, we do not see the point in diverging from the common practice of assuming knowledge of $\mathfrak{g}(\tau) \dot{\mu}(\cdot)$ and attempting to relax such an assumption in our work, given that the primary goal is not addressing heteroschedastic generalized linear bandits or heteroschedastic kernelized bandits, but combining generalized linear bandits and kernelized bandits.**
> > >
> > >         * Regarding a possible dependence on $\sqrt{\sum_t \sigma_t^2}$: we clarified above what we mean by heteroscedasticity in our setting. It is not our goal to obtain such a result, in line with the generalized linear bandit literature. By the way, the variance-related terms $\mathfrak{g}(\tau) \dot{\mu}(\cdot)$ will appear in the maximal information gain $\widetilde{\gamma}_T$.

---

> > > > ### Author Response · Authors · 2025-08-03
> > > > **Re: Official Comment by Reviewer nkzS (cont.)**
> > > >
> > > > ### References
> > > >
> > > > [i] Chowdhury, Sayak Ray, and Aditya Gopalan. "On kernelized multi-armed bandits." International Conference on Machine Learning. PMLR, 2017.
> > > >
> > > > [ii] Marc Abeille, Louis Faury, and Clément Calauzènes. Instance-wise minimax-optimal algorithms for logistic bandits. In International Conference on Artificial Intelligence and Statistics (AISTATS), pages 3691–3699. PMLR, 2021.
> > > >
> > > > [iii] Whitehouse, Justin, Aaditya Ramdas, and Steven Z. Wu. "On the sublinear regret of GP-UCB." Advances in Neural Information Processing Systems 36 (2023): 35266-35276.
> > > >
> > > > [iv] Louis Faury, Marc Abeille, Clément Calauzènes, and Olivier Fercoq. Improved optimistic algorithms for logistic bandits. In International Conference on Machine Learning (ICML), pages 3052–3060. PMLR, 2020.

---

> ### Comment · Reviewer_nkzS · 2025-08-03
>
> - I believe I have understood what the authors try to claim about the reason behind why they sometimes give a worst-case bound an executation-dependent quantity but in some other cases (in the finite-dimensional setting) are in need of manifesting that the $\log\det(...)$ quantity is not always as bad as $\mathrm{poly}(d)$.
> - If my understanding is correct, then, let us now focus on the penultimate column of Table 1 of this submission, whose title is "Heterosc."; my concerns are as follows.
>   - Following the authors' reasoning about **what the authors intended by heteroscedasticity**, then may I ask **which paper on generalized linear bandits (under assumptions similar to this submission) cannot deal with heteroscedastic feedback noise**? For example, it seems that the authors' reasoning implies that all published papers by far on logistic bandits can deal with heterscedasticity.
>   - According to the explanation by the authors on **what they intended by heteroscedasticity**, the check marks in the penultimate column of Table 1 are only because the data model of interest is generalized linear models, and the contrast between the cross marks and check marks **in this column** is not a part of the "new knowledge" introduced by this submission.

---

> > ### Author Response · Authors · 2025-08-03
> > **Re: Official Comment by Reviewer nkzS**
> >
> > * We are happy to have clarified this point.
> >
> > * Regarding the additional concerns about heteroscedasticity. First of all, **Table 1, as explained in the caption, is about the properties of the self-normalized concentration inequalities**, not the properties of the regret minimization algorithms.
> >
> >     * Clearly, the papers on generalized linear bandits *can deal with* heteroscedastic noise. The key point is whether they are able to effectively capture it in the concentration tools they use to derive tight regret bounds. As also explained in [iv, Section 3.1, paragraph "Comparison to prior work"], one could deal with heteroscedasticity even using the inequality of Abbasi-Yadkori et al. (2011) [1], simply by considering the maximum variance/sub-Gaussian proxy. However, this would introduce a dependence on the minimum variance (denoted by $\omega$ in [iv]), which is known to be suboptimal for generalized linear bandits. For instance, early works on generalized linear bandits *can deal with* heteroscedasticity, but they apply concentration tools that do not take it into account, resulting in a *suboptimal dependence on the minimum variance*, see [v, Theorem 2, the $\kappa$ term] and [vi, Theorem 2, the $c_{\mu}$ term]. Instead, [ii] and [iv], as well as our work, display a tight dependence on $\kappa_{*}$.
> >      * The penultimate column of Table 1 and the corresponding marks address the question: **"Is the concentration inequality taking into account the possible heteroscedasticity due to the exponential family noise?"** First, we believe this is a meaningful question in its own right, as a property of the concentration inequality. Second, this question also makes sense from the regret minimization perspective: as explained above, nothing prevents applying, for instance, the inequality of Abbasi-Yadkori et al. (2011) [1] while ignoring heteroscedasticity and merely considering the maximum variance/sub-Gaussian proxy, which leads to suboptimal regret bounds. Finally, our self-normalized concentration inequality allows us to answer "yes" to the above question in the setting of generalized kernelized bandits. For this reason, we consider it to represent "new knowledge" that was not previously available in the literature.
> >
> > ### References
> >
> > [v] Li, Lihong, Yu Lu, and Dengyong Zhou. "Provably optimal algorithms for generalized linear contextual bandits." International Conference on Machine Learning. PMLR, 2017.
> >
> > [vi] Filippi, Sarah, et al. "Parametric bandits: The generalized linear case." Advances in Neural Information Processing Systems 23 (2010).

---

> > > ### Comment · Reviewer_nkzS · 2025-08-04
> > >
> > > - I have carefully read the authors response on **what they intend to claim by "heterscedasticity"**.
> > > - I have re-evaluated this submission based on all the discussions by far.
> > > - I have submitted my final rating and final justification.

---

### Author Response · Authors · 2025-08-04
**Preliminary Experiment**

As requested by Reviewers 37bp and xX8K, during the rebuttal/discussion period, we conducted a preliminary experiment. Given the short time available, we are only able to provide an initial experiment comparing:

- our **GKB-UCB** algorithm, in its *efficient implementation* version described in Appendix A (due to the limited timeframe, we are unable to include a comparison with the version of the algorithm provided in the main paper);
- the **IGP-UCB** algorithm from *Chowdhury, Sayak Ray, and Aditya Gopalan. "On kernelized multi-armed bandits." International Conference on Machine Learning. PMLR, 2017*, designed for **kernelized bandits**;
- the **OFULog-r** algorithm from *Abeille, Marc, Louis Faury, and Clément Calauzènes. "Instance-wise minimax-optimal algorithms for logistic bandits." International Conference on Artificial Intelligence and Statistics. PMLR, 2021*, designed for **generalized linear bandits**.

We considered the setting of **generalized kernelized bandits** with the following configuration:

- decision space: $\mathcal{X} = [0,1]$;
- Gaussian kernel: $k(x,x') = \exp(-L(x - x')^2)$ with $L = 30$ (hence $K = 1$);
- sigmoidal link function: $\mu(x) = \frac{1}{1 + e^{-x}}$ with Bernoulli observations;
- true function:
  $$
  f(x) = 0.47 \exp(-10(x - 0.2)^2) + 0.5 \exp(-10(x - 0.8)^2) + 0.48
  $$
  This function has a global maximum close to 1 in value to emphasize the effects of the sigmoidal link function, along with a local maximum of similar value.

To run the algorithms, we discretized the decision space $\mathcal{X}$ using a regular grid of 11 points. We set $\delta = 1/T$ and $\lambda = 1 + 2/T$ (required by IGP-UCB) for all methods, and tested different time horizons: $T \in \\{10, 20, 50, 100, 200, 500\\}$.

For IGP-UCB, which requires the sub-Gaussian proxy $R$, we set $R = 1/2$, as appropriate for Bernoulli random variables. Since OFULog-r requires a finite-dimensional feature representation of the decisions, we used a Taylor expansion of the Gaussian kernel up to order 10, resulting in the following features:
$$
\phi(x) = \exp(-L x^2) \left[ 1, \sqrt{\frac{2L}{1!}} x, \sqrt{\frac{(2L)^2}{2!}} x^2, \dots, \sqrt{\frac{(2L)^{10}}{10!}} x^{10} \right]^\top
$$

The **cumulative regret** for the three algorithms is reported in the table below (as we cannot provide external links in compliance with NeurIPS policy). Each algorithm was run over 5 different seeds, and each cell reports the mean ± standard deviation.

| Method     | 10               | 20               | 50               | 100              | 200              | 500              |
|------------|------------------|------------------|------------------|------------------|------------------|------------------|
| GKB-UCB    | $0.77 \pm 0.14$  | $1.39 \pm 0.43$  | $3.06 \pm 0.69$  | $4.82 \pm 0.91$  | $8.06 \pm 1.97$  | $14.78 \pm 4.41$ |
| OFULog-r   | $0.73 \pm 0.03$  | $1.48 \pm 0.14$  | $3.70 \pm 0.45$  | $6.75 \pm 0.85$  | $13.17 \pm 0.61$ | $36.55 \pm 1.06$ |
| IGP-UCB    | $0.85 \pm 0.04$  | $1.71 \pm 0.09$  | $3.98 \pm 0.20$  | $7.41 \pm 0.41$  | $13.27 \pm 0.73$ | $27.70 \pm 2.30$ |

We observe that **GKB-UCB consistently achieves lower cumulative regret** on average than the baselines for all $T \geq 20$. Moreover, the regret exhibits a **sublinear trend**, as expected.

We are committed to integrating additional experiments in the final manuscript, and will also provide the code for the efficient implementation (using `cvxopt`), along with the code used for the baseline methods.

---

> ### Comment · Reviewer_xX8K · 2025-08-05
>
> Thanks for including the experiments! Minor comments (for camera-ready revision mostly):
> - OFULog-r (Abeille et al., 2021) is for **logistic bandits**, not generalized linear bandits
> - Can the authors consider the current SOTA logistic bandit algorithm, OFUGLB, due to Lee et al. (2024)? Their algorithm, I believe, is applicable to any generalized linear bandits as well. If the time allows for the camera-ready revision, {linear, logistic, Poisson} bandit experiments would further strengthen the paper.
> - Although it is true that the pseudocode in Appendix A is more efficient than Algorithm 1, it still requires solving the MLE each iteration. I believe a more suitable terminology is "tractable implementation." The authors should carefully include discussions such as space/time complexity. Some relevant references: [1,2,3].
>
> [1] https://arxiv.org/abs/2507.11847
>
> [2] https://openreview.net/forum?id=ofa1U5BJVJ
>
> [3] https://openreview.net/forum?id=Q4NWfStqVf

---

> > ### Author Response · Authors · 2025-08-06
> > **Re: Official Comment by Reviewer xX8K**
> >
> > Thank you for the feedback and for your support!
> >
> > - Yes, absolutely. In the rush, we wrote "generalized linear bandits," but we actually ran it with the sigmoid link function, that is, on a logistic bandit.
> >
> > - Yes, for the camera-ready version, we will run Lee et al. (2024) on the three suggested scenarios: linear, logistic, and Poisson.
> >
> > - We agree that "tractable" better describes the version of the algorithm reported in Appendix A. We will replace "efficient" with "tractable" in the camera-ready version. We will also compute the time and space complexity of this version, in line with what is done in the referenced works.

---

### Decision · Program_Chairs · 2025-09-17

**Decision:**

Reject

**Comment:**

This paper studies the problem of the generalized kernelized bandit, which combines the benefits of GLM and kernel bandits. It first aims to give self-normalized concentration in the Bernstein-Like rather than the Hoeffding-style, and then leverage it to give regret bounds. The reviews are divided among reviewers. After reading the reviews and the paper carefully myself, I think the current version is not ready for publication due to the following reasons.

1. As pointed out by the reviewer, the authors are sloppy in some parts of the proof, especially when handling infinite dimensions, which seem to be the only new things here. I would recommend that authors read the proof of Corollary 3.5 of Yasin Abbasi-Yadkori's PhD thesis (of course, the authors also need to cite it)

2. The authors seem to overclaim several contributions. I am also not quite comfortable with the claim about Theorem 6.1. A similar style result already appears in the literature a long time ago, see Lemma 3 in [R1], although it is with log rather than loglog (I think the key idea is already there)

In sum, I think the authors need to do a better job when presenting their proof and contributions by carefully reviewing the literature and giving proper credit to prior work.

---

[R1] Rakhlin, Alexander, Ohad Shamir, and Karthik Sridharan. "Making gradient descent optimal for strongly convex stochastic optimization." arXiv preprint arXiv:1109.5647 (2011).